# GEODIFFUSION: TEXT-PROMPTED GEOMETRIC CONTROL FOR OBJECT DETECTION DATA GENERATION

**Kai Chen**[1*]**, Enze Xie**[2*]**, Zhe Chen**[3]**, Yibo Wang**[4]**, Lanqing Hong**[2†]**,**
**Zhenguo Li**[2]**, Dit-Yan Yeung**[1]

[1]Hong Kong University of Science and Technology   [2]Huawei Noah's Ark Lab
[3]Nanjing University   [4]Tsinghua University

`kai.chen@connect.ust.hk`, `{xie.enze,honglanqing,li.zhenguo}@huawei.com`
`chenzhe98@smail.nju.edu.cn`, `wyb22@mails.tsinghua.edu.cn`, `dyyeung@cse.ust.hk`

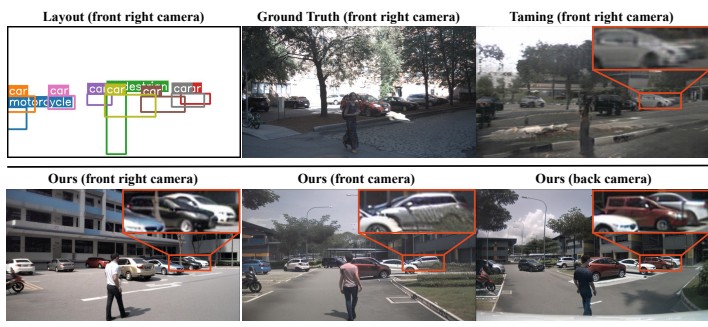
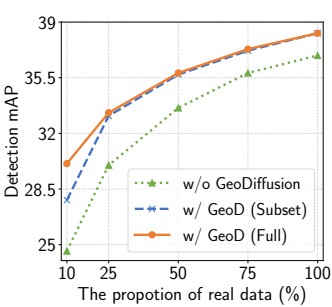

(a) **Qualitative comparison** between our GEODIFFUSION and the state-of-the-art layout-to-image (L2I) generation method (Jahn et al., 2021). See more applications (*e.g.*, **3D geometric controls**) in Appendix D-E.

(b) **Curve of mAP** with respect to the portion (%) of real data usage (*c.f.* Tab. 8).

Figure 1: (a) GEODIFFUSION can generate object detection data by encoding different geometric conditions (*e.g.*, **bounding boxes** (bottom left) and **camera views** (bottom middle & right)) with a unified architecture. (b) **For the first time**, we demonstrate that L2I-generated images can benefit the training of object detectors (Ren et al., 2015), especially under annotation-scarce circumstances.

## ABSTRACT

Diffusion models have attracted significant attention due to the remarkable ability to create content and generate data for tasks like image classification. However, the usage of diffusion models to generate the high-quality object detection data remains an underexplored area, where not only image-level perceptual quality but also geometric conditions such as bounding boxes and camera views are essential. Previous studies have utilized either copy-paste synthesis or layout-to-image (L2I) generation with specifically designed modules to encode the semantic layouts. In this paper, we propose the GEODIFFUSION, a simple framework that can flexibly translate various geometric conditions into text prompts and empower pre-trained text-to-image (T2I) diffusion models for high-quality detection data generation. Unlike previous L2I methods, our GEODIFFUSION is able to encode not only the bounding boxes but also extra geometric conditions such as camera views in self-driving scenes. Extensive experiments demonstrate GEODIFFUSION outperforms previous L2I methods while maintaining 4× training time faster. To the best of our knowledge, this is the first work to adopt diffusion models for layout-to-image generation with geometric conditions and demonstrate that L2I-generated images can be beneficial for improving the performance of object detectors.

## 1 INTRODUCTION

The cost of real data collection and annotation has been a longstanding problem in the field of deep learning. As a more cost-effective alternative, data generation techniques have been investigated for potential performance improvement (Perez & Wang, 2017; Bowles et al., 2018). However, effectiveness of such techniques has not met the expectation, mainly limited by the quality of generated data.

---

*Equal contribution. †Corresponding author.
Project Page: `https://kaichen1998.github.io/projects/geodiffusion/`.

Recently, diffusion models (DMs) (Ho et al., 2020; Nichol & Dhariwal, 2021) have emerged as one of the most popular generative models, owing to the remarkable ability to create content. Moreover, as demonstrated by He *et al.* (He et al., 2022), DMs can generate high-quality images to improve the performance of classification models. However, the usage of DMs to generate data for complex perception tasks (*e.g.*, object detection (Caesar et al., 2020; Han et al., 2021; Li et al., 2022)) has been rarely explored, which requires to consider about not only image-level perceptual quality but also geometric controls (*e.g.*, bounding boxes and camera views). Thus, there is a need to investigate how to effectively utilize DMs for generating high-quality data for such perception tasks.

Existing works primarily utilize two manners to employ generative models for *controllable* detection data generation: 1) copy-paste synthesis (Dvornik et al., 2018; Zhao et al., 2022) and 2) layout-to-image (L2I) generation (Sun & Wu, 2019; Zhao et al., 2019). Copy-paste synthesis involves by generating the foreground objects and placing them on a pre-existing background image. Although proven beneficial for detectors, it only combines different parts of images instead of generating a complete scene, leading to less realistic images. L2I generation, on the other hand, adopts classical generative models (VAE (Kingma & Welling, 2013) and GAN (Goodfellow et al., 2014)), to directly generate realistic detection data conditional on semantic layouts. However, L2I generation relies on specifically designed modules (*e.g.*, RoI Align (Zhao et al., 2019; He et al., 2021) and layout attention (Li et al., 2023b; Cheng et al., 2023)) to encode layouts, limiting its flexibility to incorporate extra geometric conditions such as camera views. Therefore, the question arises: *Can we utilize a pre-trained powerful text-to-image (T2I) diffusion model to encode various geometric conditions for high-quality detection data generation?*

Inspired by the recent advancement of language models (LMs) (Chen et al., 2023b; Gou et al., 2023), we propose GEODIFFUSION, a simple framework to translate different geometric conditions as a "foreign language" via text prompts to empower pre-trained text-to-image diffusion models (Rombach et al., 2022) for high-quality object detection data generation. Different from the previous L2I methods which can only encode bounding boxes, our work can encode various additional geometric conditions flexibly benefiting from translating conditions into text prompts (*e.g.*, GEODIFFUSION is able to control image generation conditioning on camera views in self-driving scenes). Considering the extreme imbalance among foreground and background regions, we further propose a foreground re-weighting mechanism which adaptively assigns higher loss weights to foreground regions while considering the area difference among foreground objects at the same time. Despite its simplicity, GEODIFFUSION generates highly realistic images consistent with geometric layouts, significantly surpassing previous L2I methods (**+21.85 FID** and **+27.1 mAP** compared with LostGAN (Sun & Wu, 2019) and **+12.27 FID** and **+11.9 mAP** compared with the ControlNet (Zhang et al., 2023)). **For the first time**, we demonstrate generated images of L2I models can be beneficial for training object detectors, particularly in annotation-scarce scenarios. Moreover, GEODIFFUSION can generate novel images for simulation (Fig. 4) and support complicated image inpainting requests (Fig. 5).

The main contributions of this work contain three parts:

1. We propose GEODIFFUSION, an embarrassingly simple framework to integrate geometric controls into pre-trained diffusion models for detection data generation via text prompts.

2. With extensive experiments, we demonstrate that GEODIFFUSION outperforms previous L2I methods by a significant margin while maintaining highly efficient (approximately $4\times$ training acceleration).

3. For the first time, we demonstrate that the generated images of layout-to-image models can be beneficial to training object detectors, especially for the annotation-scarce circumstances in object detection datasets.

## 2 RELATED WORK

**Diffusion models.** Recent progress in generative models has witnessed the success of diffusion models (Ho et al., 2020; Song et al., 2020), which generates images through a progressive denoising diffusion procedure starting from a normal distributed random sample. These models have shown exceptional capabilities in image generation and potential applications, including text-to-image synthesis (Nichol et al., 2021; Ramesh et al., 2022), image-to-image translation (Saharia et al., 2022a;b), inpainting (Wang et al., 2022), and text-guided image editing (Nichol et al., 2021; Hertz et al., 2022). Given the impressive success, employing diffusion models to generate perception-centric training data holds significant promise for exploiting the boundaries of perceptual model capabilities.

Table 1: **Key differences between our GEODIFFUSION and existing works.** GEODIFFUSION can generate highly realistic detection data with flexible fine-grained text-prompted geometric controls.

| Paradigm | Realistic | Layout Control | Extra Geo. Control |
|---|---|---|---|
| Copy-paste (Zhao et al., 2022) | ✗ | ✓ | ✗ |
| Layout-to-image (Li et al., 2023b) | ✓ | ✓ | ✗ |
| Perception-based (Wu et al., 2023) | ✓ | ✗ | ✗ |
| **GeoDiffusion** | ✓ | ✓ | ✓ |

**Copy-paste synthesis.** Considering that object detection models require a large amount of data, the replication of image samples, also known as copy-paste, has emerged as a straightforward way to improve data efficiency of object detection models. Nikita *et al*. (Dvornik et al., 2018) first introduce Copy-Paste as an effective augmentation for detectors. Simple Copy-Paste (Ghiasi et al., 2021) uses a simple random placement strategy and yields solid improvements. Recently, (Ge et al., 2022; Zhao et al., 2022) perform copy-paste synthesis by firstly generating foreground objects which are copied and pasted on a background image. Although beneficial for detectors, the synthesized images are: a) *not realistic*; b) *no controllable* on fine-grained geometric conditions (*e.g.*, camera views). Thus, we focus on integrating various geometric controls into pre-trained diffusion models.

**Layout-to-image generation** aims at taking a graphical input of a high-level layout and generating a corresponding photorealistic image. To address, LAMA (Li et al., 2021) designs a locality-aware mask adaption module to adapt the overlapped object masks during generation, while Taming (Jahn et al., 2021) demonstrates a conceptually simple model can outperform previous highly specialized systems when trained in the latent space. Recently, GLIGEN (Li et al., 2023b) introduces extra gated self-attention layers into pre-trained diffusion models for layout control, and LayoutDiffuse (Cheng et al., 2023) utilizes novel layout attention modules specifically designed for bounding boxes. The most similar with ours is ReCo (Yang et al., 2023), while GEODIFFUSION further 1) supports extra geometric controls purely with text prompts, 2) proposes the foreground prior re-weighting for better foreground object modeling and 3) demonstrates L2I-generated images can benefit object detectors.

**Perception-based generation.** Instead of conducting conditional generation given a specific input layout, perception-based generation aims at generating corresponding annotations simultaneously during the original unconditional generation procedure by adopting a perception head upon the pre-trained diffusion models. DatasetDM (Wu et al., 2023) learns a Mask2Former-style P-decoder upon a fixed Stable Diffusion model, while Li *et al*. (Li et al., 2023c) further propose a fusion module to support open-vocabulary segmentation. Although effective, perception-based methods 1) can hardly outperform directly combining pre-trained diffusion models with specialized open-world perception models (*e.g.*, SAM (Kirillov et al., 2023)), 2) solely rely on pre-trained diffusion models for image generation and cannot generalize to other domains (*e.g.*, driving scenes) and 3) only support textual-conditional generation, neither the fine-grained geometric controls (*e.g.*, bounding boxes and camera views) nor sophisticated image editing requests (*e.g.*, inpainting).

## 3 METHOD

In this section, we first introduce the basic formulation of our *generalized layout-to-image* (GL2I) generation problem with geometric conditions and diffusion models (DMs) (Ho et al., 2020) in Sec. 3.1.1 and 3.1.2 separately. Then, we discuss how to flexibly encode the geometric conditions via text prompts to utilize pre-trained text-to-image (T2I) diffusion models (Rombach et al., 2022) and build our GEODIFFUSION in Sec. 3.2 and 3.3.

### 3.1 PRELIMINARY

#### 3.1.1 GENERALIZED LAYOUT-TO-IMAGE GENERATION

Let $L = (v, \{(c_i, b_i)\}_{i=1}^N)$ be a *geometric layout* with $N$ bounding boxes, where $c_i \in \mathcal{C}$ denotes the semantic class, and $b_i = [x_{i,1}, y_{i,1}, x_{i,2}, y_{i,2}]$ represents locations of the bounding box (*i.e.*, top-left and bottm-right corners). $v \in \mathcal{V}$ can be any extra *geometric conditions* associated with the layout. Without loss of generality, we take camera views as an example in this paper. Thus, the *generalized layout-to-image* generation aims at learning a $\mathcal{G}(\cdot, \cdot)$ to generate images $I \in \mathcal{R}^{H \times W \times 3}$ conditional on given geometric layouts $L$ as $I = \mathcal{G}(L, z)$, where $z \sim \mathcal{N}(0, 1)$ is a random Gaussian noise.

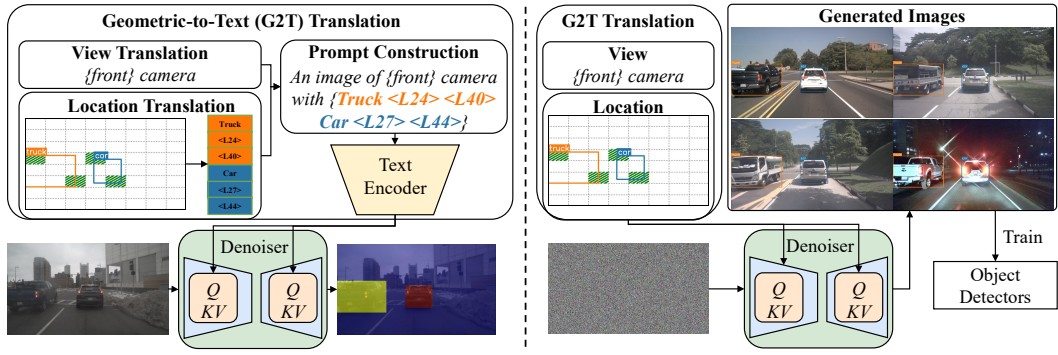

Figure 2: **Model architecture of GEODIFFUSION.** (a) GEODIFFUSION encodes various geometric conditions via text prompts to empower text-to-image (T2I) diffusion models for generalized layout-to-image generation with various geometric conditions, even supporting the 3D geometric conditions as shown in Fig. 11. (b) GEODIFFUSION can generate highly realistic and diverse detection data to benefit the training of object detectors.

### 3.1.2 CONDITIONAL DIFFUSION MODELS

Different from typical generative models like GAN (Goodfellow et al., 2014) and VAE (Kingma & Welling, 2013), diffusion models (Ho et al., 2020) learn data underlying distribution by conducting a $T$-step denoising process from normally distributed random variables, which can also be considered as learning an inverse process of a fixed Markov Chain of length $T$. Given a noisy input $x_t$ at the time step $t \in \{1, ..., T\}$, the model $\epsilon_\theta(x_t, t)$ is trained to recover its clean version $x$ by predicting the random noise added at time step $t$, and the objective function can be formulated as,

$$\mathcal{L}_{DM} = \mathbb{E}_{x,\epsilon \sim \mathcal{N}(0,1),t} \| \epsilon - \epsilon_\theta(x_t, t) \|^2. \tag{1}$$

Latent diffusion models (LDM) (Rombach et al., 2022) instead perform the diffusion process in the latent space of a pre-trained Vector Quantized Variational AutoEncoder (VQ-VAE) (Van Den Oord et al., 2017). The input image $x$ is first encoded into the latent space of VQ-VAE encoder as $z = \mathcal{E}(x) \in \mathcal{R}^{H' \times W' \times D'}$, and then taken as clean samples in Eqn. 1. To facilitate conditional generation, LDM further introduces a conditional encoder $\tau_\theta(\cdot)$, and the objective can be formulated as,

$$\mathcal{L}_{LDM} = \mathbb{E}_{\mathcal{E}(x),\epsilon \sim \mathcal{N}(0,1),t} \| \epsilon - \epsilon_\theta(z_t, t, \tau_\theta(y)) \|^2, \tag{2}$$

where $y$ in the introduced condition (*e.g.*, text in LDM (Rombach et al., 2022)).

### 3.2 GEOMETRIC CONDITIONS AS A FOREIGN LANGUAGE

In this section, we explore encoding various geometric conditions via text prompts to utilize the pre-trained text-to-image diffusion models for better GL2I generation. As discussed in Sec. 3.1.1, a geometric layout $L$ consists of three basic components, including the locations $\{b_i\}$ and the semantic classes $\{c_i\}$ of bounding boxes and the extra geometric conditions $v$ (*e.g.*, camera views).

**Location "Translation".** While classes $\{c_i\}$ and conditions $v$ can be naturally encoded by replacing with the corresponding textual explanations (see Sec. 4.1 for details), locations $\{b_i\}$ can not since the coordinates are continuous. Inspired by (Chen et al., 2021b), we discretize the continuous coordinates by dividing the image into a grid of **location bins**. Each location bin corresponds to a unique **location token** which will be inserted into the text encoder vocabulary of diffusion models. Therefore, each *corner* can be represented by the *location token* corresponding to the *location bin* it belongs to, and the "translation" procedure from the box locations to plain text is accomplished. See Fig. 2 for an illustration. Specifically, given a grid of size $(H_{bin}, W_{bin})$, the corner $(x_0, y_0)$ is represented by the $\sigma(x_0, y_0)$ location token as,

$$\sigma(x_0, y_0) = \mathcal{T}[y_{bin} * W_{bin} + x_{bin}], \tag{3}$$

$$x_{bin} = \lfloor x_0/W * W_{bin} \rfloor, \ y_{bin} = \lfloor y_0/H * H_{bin} \rfloor, \tag{4}$$

where $\mathcal{T} = \{< L_i >\}_{i=1}^{H_{bin} * W_{bin}}$ is the set of location tokens, and $\mathcal{T}[\cdot]$ is the index operation. Thus, a bounding box $(c_i, b_i)$ is encoded into a "pharse" with three tokens as $(c_i, \sigma(x_{i,1}, y_{i,1}), \sigma(x_{i,2}, y_{i,2}))$.

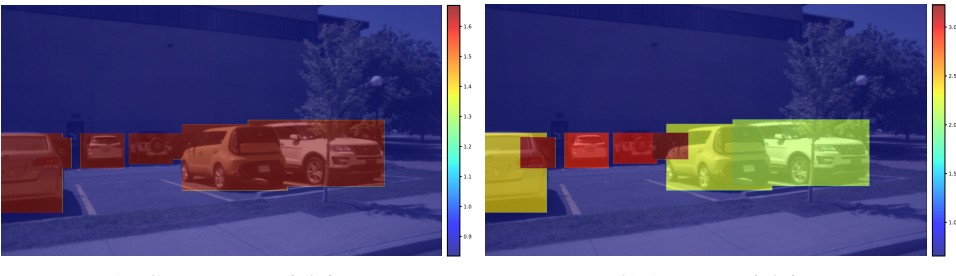

(a) Constant re-weighting.      (b) Area re-weighting.

Figure 3: **Foreground prior re-weighting**. (a) Constant re-weighting assigns equal weight to all the bounding boxes, while (b) area re-weighting adaptively assigns higher weights to smaller boxes.

**Prompt construction.** To generate a text prompt, we can serialize multiple box "phrases" into a single sequence. In line with (Chen et al., 2021b), here we utilize a random ordering strategy by randomly shuffling the box sequence each time a layout is presented to the model. Finally, we construct the text prompts with the template, "An image of {view} camera with {boxes}". As demonstrated in Tab. 2, this seemingly simple approach can effectively empower the T2I diffusion models for high-fidelity GL2I generation, and even *3D geometric controls* as shown in Fig. 11.

### 3.3 FOREGROUND PRIOR RE-WEIGHTING

The objective function presented in Eqn. 2 is designed under the assumption of a uniform prior distribution across spatial coordinates. However, due to the extreme imbalance between foreground and background, we further introduce an adaptive re-weighting mask, denoted by $m \in \mathcal{R}^{H' \times W'}$, to adjust the prior. This enables the model to focus more thoroughly on foreground generation and better address the challenges posed by the foreground-background imbalance.

**Constant re-weighting.** To distinguish the foreground from background regions, we employ a re-weighting strategy whereby foreground regions are assigned a weight $w(w > 1)$, greater than that assigned to the background regions.

**Area re-weighting.** The constant re-weighting strategy assigns equal weight to all foreground boxes, which causes larger boxes to exert a greater influence than smaller ones, thereby hindering the effective generation of small objects. To mitigate this issue, we propose an area re-weighting method to dynamically assign higher weights to smaller boxes. A comparison of this approach can be seen in Fig. 3. Finally, the re-weighting mask $m$ can be represented as,

$$m'_{ij} = \begin{cases} w/c^p_{ij} & (i,j) \in \text{foreground} \\ 1/(H' * W')^p & (i,j) \in \text{background} \end{cases}, \tag{5}$$

$$m_{ij} = H' * W' * m'_{ij}/\sum m'_{ij}, \tag{6}$$

where $c_{ij}$ represents the area of the bounding box to which pixel $(i,j)$ belongs, and $p$ is a tunable parameter. To improve the numerical stability during the fine-tuning process, Eqn. 6 normalizes the re-weighting mask $m'$. The benefits of this normalization process are demonstrated in Tab. 9.

**Objective function.** The final objective function of our GEODIFFUSION can be formulated as,

$$\mathcal{L}_{\text{GeoDiffusion}} = \mathbb{E}_{\mathcal{E}(x),\epsilon,t} \|\epsilon - \epsilon_\theta(z_t, t, \tau_\theta(y))\|^2 \odot m, \tag{7}$$

where $y$ is the encoded layout as discussed in Sec. 3.2.

## 4 EXPERIMENTS

### 4.1 IMPLEMENTATION DETAILS

**Dataset.** Our experiments primarily utilize the widely used NuImages (Caesar et al., 2020) dataset, which consists of 60K training samples and 15K validation samples with high-quality bounding box annotations from 10 semantic classes. The dataset captures images from 6 camera views (*front, front left, front right, back, back left and back right*), rendering it a suitable choice for our GL2I generation problem. Moreover, to showcase the universality of GEODIFFUSION for common layout-to-image settings, we present experimental results on COCO (Lin et al., 2014; Caesar et al., 2018) in Sec. 4.3.

Table 2: **Comparison of generation fidelity on NuImages.** 1) Effectiveness: GEODIFFUSION surpasses all baselines by a significant margin, suggesting the effectiveness of adopting text-prompted geometric control. 2) Efficiency: Our GEODIFFUSION generates highly-discriminative images even under annotation-scarce circumstances. "Const." and "ped." suggests construction and pedestrian separately. *: represents the real image *Oracle* baseline.

| Method | Input Res. | Ep. | FID↓ | Average Precision↑ | | | | | | | | |
|---|---|---|---|---|---|---|---|---|---|---|---|---|
| | | | | mAP | $AP_{50}$ | $AP_{75}$ | $AP^m$ | $AP^l$ | trailer | const. | ped. | car |
| Oracle* | - | - | - | 48.2 | 75.0 | 52.0 | 46.7 | 60.5 | 17.8 | 30.9 | 48.5 | 64.9 |
| LostGAN | 256×256 | 256 | 59.95 | 4.4 | 9.8 | 3.3 | 2.1 | 12.3 | 0.3 | 1.3 | 2.7 | 12.2 |
| LAMA | 256×256 | 256 | 63.85 | 3.2 | 8.3 | 1.9 | 2.0 | 9.4 | 1.4 | 1.0 | 1.3 | 8.8 |
| Taming | 256×256 | 256 | 32.84 | 7.4 | 19.0 | 4.8 | 2.8 | 18.8 | 6.0 | 4.0 | 3.0 | 17.3 |
| ReCo | 512×512 | 64 | 27.10 | 17.1 | 41.1 | 11.8 | 10.9 | 36.2 | 8.0 | 11.8 | 7.6 | 31.8 |
| GLIGEN | 512×512 | 64 | 16.68 | 21.3 | 42.1 | 19.1 | 15.9 | 40.8 | 8.5 | 14.3 | 14.7 | 38.7 |
| ControlNet | 512×512 | 64 | 23.26 | 22.6 | 43.9 | 20.7 | 17.3 | 41.9 | 10.5 | 15.1 | 16.7 | 40.7 |
| **GeoDiffusion** | 256×256 | 64 | 14.58$^{+18.26}$ | 15.6$^{+8.2}$ | 31.7 | 13.4 | 6.3 | 38.3 | 13.3 | 10.8 | 6.5 | 26.3 |
| **GeoDiffusion** | 512×512 | 64 | **9.58**$^{+23.26}$ | 31.8$^{+24.4}$ | 62.9 | 28.7 | 27.0 | 53.8 | 21.2 | 27.8 | 18.2 | 46.0 |
| **GeoDiffusion** | 800×456 | 64 | 10.99$^{+21.85}$ | **34.5**$^{+27.1}$ | **68.5** | **30.6** | **31.1** | **54.6** | **20.4** | **29.3** | **21.6** | **48.8** |

**Optimization.** We initialize the embedding matrix of the location tokens with 2D sine-cosine embeddings (Vaswani et al., 2017), while the remaining parameters of GEODIFFUSION are initialized with Stable Diffusion (v1.5), a pre-trained text-to-image diffusion model based on LDM (Rombach et al., 2022). With VQ-VAE fixed, we fine-tune all parameters of the text encoder and U-Net using AdamW (Loshchilov & Hutter, 2019) optimizer and a cosine learning rate schedule with a linear warm-up of 3000 iterations. The batch size is set to 64, and learning rates are set to $4e^{-5}$ for U-Net and $3e^{-5}$ for the text encoder. Layer-wise learning rate decay (Clark et al., 2020) is further adopted for the text encoder, with a decay ratio of 0.95. With 10% probability, the text prompt is replaced with a *null* text for unconditional generation. We fine-tune our GEODIFFUSION for 64 epochs, while baseline methods are trained for 256 epochs to maintain a similar training budget with the COCO recipe in (Sun & Wu, 2019; Li et al., 2021; Jahn et al., 2021). Over-fitting is observed if training baselines longer. During generation, we sample images using the PLMS (Liu et al., 2022a) scheduler for 100 steps with the classifier-free guidance (CFG) set as 5.0.

## 4.2 MAIN RESULTS

The quality of object detection data is predicated on three key criteria: the **fidelity**, **trainability**, and **generalizability**. Fidelity demands a realistic object representation while consistent with geometric layouts. Trainability concerns the usefulness of generated images for the training of object detectors. Generalizability demands the capacity to simulate uncollected, novel scenes in real datasets. In this section, we conduct a comprehensive evaluation of GEODIFFUSION for these critical areas.

### 4.2.1 FIDELITY

**Setup.** We utilize two primary metrics on the NuImages validation set to evaluate Fidelity. The perceptual quality is measured with the Frechet Inception Distance (FID)[1] (Heusel et al., 2017), while consistency between generated images and geometric layouts is evaluated via reporting the COCO-style average precision (Lin et al., 2014) using a pre-trained object detector, which is similar to the YOLO score in LAMA (Li et al., 2021). A Mask R-CNN[2] (He et al., 2017) model pre-trained on the NuImages training set is used to make predictions on generated images. These predictions are subsequently compared with the corresponding ground truth annotations. We further provide the detection results on real validation images as the *Oracle* baseline in Tab. 2 for reference.

**Discussion.** As in Tab. 2, GEODIFFUSION surpasses all the baselines in terms of perceptual quality (FID) and layout consistency (mAP) with 256×256 input, accompanied with a **4× acceleration** (64 vs. 256 epochs), which indicates that the text-prompted geometric control is an effective approach. Moreover, the simplicity of LDM empowers GEODIFFUSION to generate higher-resolution images with minimal modifications. With 800×456 input, GEODIFFUSION gets **10.99 FID** and **34.5 mAP**,

---

[1]Images are all resize into $256 \times 256$ before evaluation.

[2]https://github.com/open-mmlab/mmdetection3d/tree/master/configs/nuimages

Table 3: **Comparison of generation trainability on NuImages.** 1) GEODIFFUSION is **the only** layout-to-image method showing consistent improvements over almost all classes, 2) especially on rare ones, verifying that GEODIFFUSION indeed helps relieve annotation scarcity during detector training. By default the 800×456 GEODIFFUSION variant is utilized for all trainability evaluation.

| Method | mAP | car | truck | trailer | bus | const. | bicycle | motorcycle | ped. | cone | barrier |
|---|---|---|---|---|---|---|---|---|---|---|---|
| Real only | 36.9 | 52.9 | 40.9 | 15.5 | 42.1 | 24.0 | 44.7 | 46.7 | **31.3** | **32.5** | 38.9 |
| LostGAN | $35.6^{-1.3}$ | 51.7 | 39.6 | 12.9 | 41.3 | 22.7 | 42.4 | 45.6 | 30.0 | 31.6 | 37.9 |
| LAMA | $35.6^{-1.3}$ | 51.7 | 39.2 | 14.3 | 40.5 | 22.9 | 43.2 | 44.9 | 30.0 | 31.3 | 38.3 |
| Taming | $35.8^{-1.1}$ | 51.9 | 39.3 | 14.7 | 41.1 | 22.4 | 43.1 | 45.4 | 30.4 | 31.6 | 38.1 |
| ReCo | $36.1^{-0.8}$ | 52.2 | 40.9 | 14.3 | 41.8 | 24.2 | 42.8 | 45.9 | 29.5 | 31.2 | 38.3 |
| GLIGEN | $36.3^{-0.6}$ | 52.8 | 40.7 | 14.1 | 42.0 | 23.8 | 43.5 | 46.2 | 30.2 | 31.7 | 38.4 |
| ControlNet | $36.4^{-0.5}$ | 52.8 | 40.5 | 13.6 | 42.1 | 24.1 | 43.9 | 46.1 | 30.3 | 31.8 | 38.6 |
| **GeoDiffusion** | $\mathbf{38.3}^{+1.4}$ | **53.2** | **43.8** | **18.3** | **45.0** | **27.6** | **45.3** | **46.9** | 30.5 | 32.1 | **39.8** |

marking a significant progress towards bridging the gap with real images, especially for the large object generation (54.6 vs. 60.5 $AP^l$). Fig. 1a provides a qualitative comparison of generated images. GEODIFFUSION generates images that are more realistic and tightly fitting to the bounding boxes, making it feasible to enhance object detector training, as later discussed in Sec. 4.2.2.

We further report the class-wise AP of the top-2 frequent (*i.e.*, car and pedestrian) and rare classes (*e.g.*, trailer and construction, occupying only 1.5% of training annotations) in Tab. 2. We observe that GEODIFFUSION performs relatively well in annotation-scarce scenarios, achieving higher trailer AP even than the *Oracle* baseline and demonstrating ability to generate highly-discriminative objects even with limited annotations. However, similar to previous methods, GEODIFFUSION encounters difficulty with high variance (*e.g.*, pedestrians) and occlusion (*e.g.*, cars) circumstances, highlighting the areas where further improvements are still needed.

### 4.2.2 TRAINABILITY

In this section, we investigate the potential benefits of GEODIFFUSION-generated images for object detector training. To this end, we utilize the generated images as augmented samples during the training of an object detector to further evaluate the efficacy of our proposed model.

**Setup.** Taking data annotations of the NuImages training set as input, we first filter bounding boxes smaller than 0.2% of the image area, then augment the bounding boxes by randomly flipping with 0.5 probability and shifting no more than 256 pixels. The generated images are considered augmented samples and combined with the real images to expand the training data. A Faster R-CNN (Ren et al., 2015) initialized with ImageNet pre-trained weights is then trained using the standard 1× schedule and evaluated on the validation set.

**Discussion.** As reported in Tab. 3, **for the first time**, we demonstrate that the generated images of layout-to-image models can be advantageous to object detector training. Our GEODIFFUSION is the only method to achieve a consistent improvement for almost all semantic classes, which is much more obvious for rare classes (*e.g.*, **+2.8** for trailer, **+3.6** for construction and **+2.9** for bus, the top-3 most rare classes occupying only 7.2% of the training annotations), revealing that GEODIFFUSION indeed contributes by relieving annotation scarcity of rare classes, thanks to the data efficiency as discussed in Sec. 4.2.1.

**Necessity of real data** brought by GEODIFFUSION is further verified by varying the amount of real data usage. We randomly sample 10%, 25%, 50%, and 75% of the real training set, and each subset is utilized to train a Faster R-CNN together with generated images separately. We consider two augmentation modes, 1) *Full*: GEODIFFUSION trained on the full training set is utilized to augment each subset as in Tab. 3. 2) *Subset*: we re-train GEODIFFUSION with each real data subset separately, which are then used to augment the corresponding subset. The number of gradient steps are maintained unchanged for each pair experiment with the same amount of real data by enlarging the batch size adaptively. As shown in Fig. 1b, GEODIFFUSION achieves consistent improvement over different real training data budgets. The more scarce real data is, the more significant improvement GEODIFFUSION achieves, as in Tab. 8, sufficiently revealing that generated images do help ease the data necessity. With only 75% of real data, the detector can perform comparably with the full real dataset by augmenting with GEODIFFUSION.

| Real w/ Query Layout | Generated w/ Query Layout | Generated w/ Flipped Layout | Generated w/ Shifted Layout |

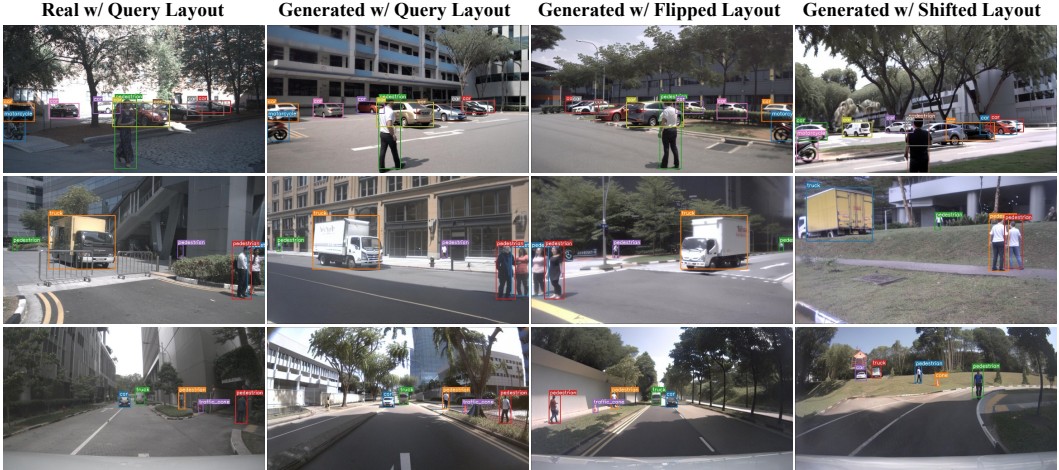

Figure 4: **Visualization of generation generalizability on NuImages**. From left to right, we present the query layout, the generated images conditional on the original, the flipped and shifted layouts. GEODIFFUSION performs surprisingly well on the real-life collected geometric layouts (2nd & 3rd columns), while revealing superior robustness for out-of-distribution situations (4th column).

### 4.2.3 GENERALIZABILITY

In this section, we evaluate the generalizability and robustness of GEODIFFUSION on novel layouts unseen during fine-tuning.

**Setup.** To guarantee that the input geometric layouts are reasonable (*e.g.*, no violation of the basic physical laws like objects closer to the camera seem larger), we first randomly sample a query layout from the validation set, based on which we further disturb the query bounding boxes with 1) *flipping* and 2) *random shifting* for no more than 256 pixels along each spatial dimension, the exact same recipe we utilze in Sec. 4.2.2 for bounding box augmentation. Check more generalizability analysis for OoD circumstances in Appendix C.

**Discussion.** We visualize the generated results in Fig. 4. GEODIFFUSION demonstrates superior generalizability to conduct generation on the novel unseen layouts. Specifically, GEODIFFUSION performs surprisingly well given geometric layouts collected in real-world scenarios and its corresponding flip variant (2nd & 3rd columns in Fig. 4). Moreover, we observe GEODIFFUSION demonstrates strong robustness to layout augmentation even if the resulting layouts are out-of-distribution. For example, GEODIFFUSION learns to generate a downhill for boxes lower than current grounding plane (*e.g.*, the *shift* column of the 1st row), or an uphill for bounding boxes higher than the current grounding plane (*e.g.*, *shift* of 2nd & 3rd rows) to maintain consistent with given geometric layouts. The remarkable robustness also convinces us to adopt bounding box augmentation for the object detector training in Sec. 4.2.2 to further increase annotation diversity of the augmented training set.

### 4.3 UNIVERSALITY

**Setup.** To demonstrate the universality of GEODIFFUSION, we further conduct experiments on COCO-Stuff dataset (Caesar et al., 2018) following common practices (Li et al., 2021; 2023b). We keep hyper-parameters consistent with Sec. 4.1, except we utilize the DPM-Solver (Lu et al., 2022) scheduler for 50-step denoising with the classifier-free guidance ratio set as 4.0 during generation.

**Fidelity.** We ignore object annotations covering less than 2% of the image area, and only images with 3 to 8 objects are utilized during validation following Li et al. (2021). Similarly with Sec. 4.2.1, we report FID and YOLO score (Li et al., 2021) to evaluate perceptual quality and layout consistency respectively. As shown in Tab. 4, GEODIFFUSION outperforms all baselines in terms of both the FID and YOLO score with significant efficiency, consistent with our observation on NuImages in Tab. 2, revealing the universality of GEODIFFUSION. More qualitative comparison is provided in Fig. 12.

**Trainability.** We then utilize GEODIFFUSION to augment COCO detection training set following the exact same box augmentation pipeline in Sec. 4.2.2. As demonstrated in Tab. 5, GEODIFFUSION also achieves significant improvement on COCO validation set, suggesting that GEODIFFUSION can indeed generate high-quality detection data regardless of domains.

Table 4: **Comparison of generation fidelity on COCO.** †: our re-implementation.

| Method | Epoch | FID↓ | mAP↑ | AP$_{50}$↑ | AP$_{75}$↑ |
|---|---|---|---|---|---|
| *256×256* | | | | | |
| LostGAN | 200 | 42.55 | 9.1 | 15.3 | 9.8 |
| LAMA | 200 | 31.12 | 13.4 | 19.7 | 14.9 |
| CAL2IM | 200 | 29.56 | 10.0 | 14.9 | 11.1 |
| Taming | 68+60 | 33.68 | - | - | - |
| TwFA | 300 | 22.15 | - | 28.2 | 20.1 |
| Frido | 200 | 37.14 | 17.2 | - | - |
| L.Diffusion† | 180 | 22.65 | 14.9 | 27.5 | 14.9 |
| **GeoDiffusion** | **60** | **20.16** | **29.1** | **38.9** | **33.6** |
| *512×512* | | | | | |
| ReCo† | 100 | 29.69 | 18.8 | 33.5 | 19.7 |
| ControlNet† | 60 | 28.14 | 25.2 | **46.7** | 22.7 |
| L.Diffuse† | 60 | 22.20 | 11.4 | 23.1 | 10.1 |
| GLIGEN | 86 | 21.04 | 22.4 | 36.5 | 24.1 |
| **GeoDiffusion** | **60** | **18.89** | **30.6** | 41.7 | **35.6** |

Table 5: **Comparison of trainability on COCO.** Indeed, GEODIFFUSION can benefit detection training regardless of domains.

| Method | mAP | AP$_{50}$ | AP$_{75}$ | AP$^m$ | AP$^l$ |
|---|---|---|---|---|---|
| Real only | 37.3 | 58.2 | 40.8 | 40.7 | 48.2 |
| L.Diffusion | 36.5 | 57.0 | 39.5 | 39.7 | 47.5 |
| L.Diffuse | 36.6 | 57.4 | 39.5 | 40.0 | 47.4 |
| GLIGEN | 36.8 | 57.6 | 39.9 | 40.3 | 47.9 |
| ControlNet | 36.9 | 57.8 | 39.6 | 40.4 | 49.0 |
| **GeoDiffusion** | **38.4** | **58.5** | **42.4** | **42.1** | **50.3** |

Table 6: **Comparison of COCO inpainting.**

| Method | mAP | AP$_{50}$ | AP$_{75}$ |
|---|---|---|---|
| Stable Diffusion | 17.6 | 23.2 | 20.0 |
| ControlNet | 17.8 | 25.7 | 20.2 |
| GLIGEN | 18.3 | 25.8 | 20.9 |
| **GeoDiffusion** | **19.0**$^{+1.4}$ | **26.2**$^{+3.0}$ | **21.6**$^{+1.6}$ |

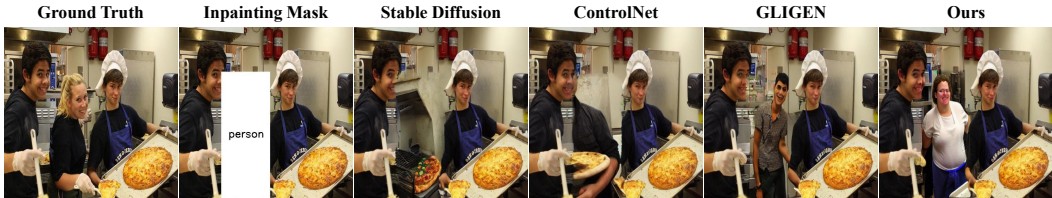

| Ground Truth | Inpainting Mask | Stable Diffusion | ControlNet | GLIGEN | Ours |
|---|---|---|---|---|---|

Figure 5: **Visualization of image inpainting on COCO.** Due to the existence of multiple people, Stable Diffusion cannot properly deal with the inpainting request, while GEODIFFUSION solves by successfully understanding location of missing areas thanks to the text-prompted geometric control.

**Inpainting.** We further explore applicability of inpainting for GEODIFFUSION. Similarly with the fidelity evaluation, we randomly mask one object for each image of COCO detection validation set, and ask models to inpaint missing areas. A YOLO detector is adopted to evaluate recognizability of inpainted results similarly with YOLO score following Li et al. (2023b). GEODIFFUSION surpasses SD baseline remarkably, as in Tab. 6. We further provide a qualitative comparison in Fig. 5, where GEODIFFUSION can successfully deal with the sophisticated image synthesis requirement.

## 4.4 ABLATION STUDY

In this section, we conduct ablation studies on the essential components of our GEODIFFUSION. Check detailed experiment setups, more ablations and discussions in Tab. 9-10 and Appendix B-D.

**Location grid size** $(H_{bin}, W_{bin})$ is ablated in Tab. 7. A larger grid size can achieve consistent improvement for both the perceptual quality and the layout consistency. Indeed, a larger grid size stands for a smaller bin size and less coordinate

Table 7: **Ablations on location grid size.** Foreground re-weighting is not adopted here. Default settings are marked in gray.

| Grid size $(H_{bin}, W_{bin})$ | # Pixels / bin | FID↓ | mAP↑ |
|---|---|---|---|
| 100×57 | 8×8 | 14.94 | 20.8 |
| 200×114 | 4×4 | 11.83 | 21.4 |
| **400×228** | **2×2** | **11.63** | **23.7** |

discretization error, and thus, a more accurate encoding of geometric layouts. Due to the restriction of hardware resources, the most fine-grained grid size we adopt is 2×2 pixels per bin.

## 5 CONCLUSION

This paper proposes GEODIFFUSION, an embarrassingly simple architecture with the text-prompted geometric control to empower pre-trained text-to-image diffusion models for object detection data generation. GEODIFFUSION is demonstrated effective in generating realistic images that conform to specified geometric layouts, as evidenced by the extensive experiments that reveal high fidelity, enhanced trainability in annotation-scarce scenarios, and improved generalizability to novel scenes.

**Acknowledgement.** We gratefully acknowledge the support of the MindSpore, CANN (Compute Architecture for Neural Networks) and Ascend AI Processor used for this research. This research has been made possible by funding support from the Research Grants Council of Hong Kong through the Research Impact Fund project R6003-21.

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

Table 8: **Necessity of real data.** GEODIFFUSION achieves consistent improvement over various real data budget, which is more significant on more annotation-scarce subsets.

| Method | 10% | 25% | 50% | 75% |
|---|---|---|---|---|
| w/o GeoDiffusion | 24.6 | 30.0 | 33.6 | 35.8 |
| **w/ GeoDiffusion (Subset)** | 27.8$^{+3.2}$ | 33.1$^{+3.1}$ | 35.7$^{+2.1}$ | 37.2$^{+1.4}$ |
| **w/ GeoDiffusion (Full)** | **30.1**$^{+5.5}$ | **33.3**$^{+3.3}$ | **35.8**$^{+2.2}$ | **37.3**$^{+1.5}$ |

## APPENDIX

## A MORE EXPERIMENTS

**Detailed results of real data necessity.** As discussed in Sec. 4.2.2, the usage of the augmented dataset generated by GEODIFFUSION can significantly ease the necessity of real data during object detector training in different real training data budget ranging from 10% to 75%. We provide detailed experimental results in Tab. 8.

## B MORE ABLATION STUDY

**Setup.** We conduct ablation studies mainly with respect to fidelity and report the FID and COCO-style mAP following the exact same setting in Sec. 4.2.1. Specifically, the input resolution is set to be 800×456, and our GEODIFFUSION is fine-tuned for 64 epochs on the NuImages training set. The optimization recipe is maintained the same with Sec. 4.1.

**Pre-trained text encoder.** To verify the necessity of using the pre-trained text encoder, we only initialize the VQ-VAE and U-Net with Stable Diffusion, while randomly initializing the parameters of the text encoder for comparison following the official implementation of LDM. As demonstrated in Tab. 9, the default GEODIFFUSION significantly surpasses the variant without the pre-trained text encoder by 4.82 FID and 14.2 mAP, suggesting that with a proper "translation", the pre-trained text encoder indeed possesses transferability to encode geometric conditions and enable T2I diffusion models for high-quality object detection data generation.

**Foreground prior re-weighting.** In Tab. 9, we study the effect of foreground re-weighting. Adopting the constant re-weighting obtains a significant +3.4 mAP improvement (27.1 *vs.* 23.7), which further increases to 30.1 mAP with the help of area re-weighting, revealing that foreground modeling is essential for object detection data generation. Note that the mAP improvement comes at a cost of a minor FID increase (11.99 *vs.* 11.63) since we manually adjust the prior distribution over spatial locations. We further verify the effectiveness of mask normalization in Eqn. 6, which can significantly decrease FID while maintaining the mAP value almost unchanged (5th & 7th rows), suggesting that mask normalization is mainly beneficial for fine-tuning the diffusion models after foreground re-weighting.

**Importance of camera views.** In this work, camera views are considered as an example to demonstrate that text can be indeed used as a universal encoding for various geometric conditions. A toy example is provided in Fig. 1a, fully proving text indeed has the potential to decouple various conditions with a unified representation. Moreover, we further build a GEODIFFUSION *w/o camera views*, significantly worse than the default GEODIFFUSION as shown in Tab. 10, revealing the importance of adopting camera views. Check Fig. 9 for more qualitative comparison.

**Location tokens.** As stated in Sec. 4.1, location tokens are first initialized with 2D sine-cosine embeddings and then fine-tuned together with the whole model. We further train a GEODIFFUSION *with fixed location tokens*, which performs worse than the default GEODIFFUSION as in Tab. 10.

| Pre-trained text encoder | Constant $w$ | Area $p$ | Norm | FID↓ | mAP↑ |
|:---:|:---:|:---:|:---:|:---:|:---:|
| ✗ | 1.0 (no re-weight) | 0 | ✓ | 16.45 | 9.5 |
| ✓ | 1.0 (no re-weight) | 0 | ✓ | **11.63** | 23.7 |
| ✓ | 2.0 | 0 | ✓ | 11.77 | 27.1 |
| ✓ | 4.0 | 0 | ✓ | 12.90 | 26.2 |
| ✓ | **2.0** | **0.2** | ✓ | 11.99 | **30.1** |
| ✓ | 2.0 | 0.4 | ✓ | 14.99 | 28.3 |
| ✓ | 2.0 | 0.2 | ✗ | 14.76 | 29.8 |

Table 9: **Ablations on the text encoder and foreground re-weighting.** The best result is achieved when both constant and area re-weighting are adopted. Default settings are marked in gray .

| Camera views | Learned Location Token | FID↓ | mAP↑ |
|:---:|:---:|:---:|:---:|
| ✓ | ✓ | **10.99** | **34.5** |
| ✗ | ✓ | 11.95 | 31.0 |
| ✓ | ✗ | 14.36 | 27.7 |

Table 10: **Ablations on camera views and location tokens.** Default settings are marked in gray .

## C   MORE DISCUSSIONS

**More generalizability analysis.**   We first provide more visualization for augmented bounding boxes similar to Sec. 4.2.3. As shown in Fig. 6, GEODIFFUSION demonstrates superior robustness towards the real-life collected and augmented layouts, convincing us to flexibly perform bounding box augmentation for the more diverse augmented set.

We further explore the generalizability for totally out-of-distribution (OoD) layouts (*i.e.*, unseen boxes and classes) in Fig. 7. GEODIFFUSION performs surprisingly well for OoD bounding boxes (*e.g.*, unusually large bounding boxes with only one object in an entire image) as in Fig. 7a, but still suffers from unseen classes as in Fig. 7b, probably due to the inevitable forgetting during fine-tuning. Parameter-efficient fine-tuning (PEFT) (Cheng et al., 2023; Li et al., 2023b) might ease the problem but at a cost of generation quality as shown in Tab. 4. Considering our focus is to generate high-quality detection data to augment real data, fidelity, and trainability are considered as the primary criteria in this work.

**Advantage over ControlNet**   mainly lies in the simplicity of GEODIFFUSION emerging especially when extended to multi-conditional generation, where usually more than 3 conditions are considered for a single generative model simultaneously (*e.g.*, 3D geometric controls (Gao et al., 2023), multi-object tracking (Li et al., 2023a) and the implicit concept removal (Liu et al., 2023)). Different from ControlNet (Zhang et al., 2023) requiring *different copies of parameters for different conditions*, our GEODIFFUSION utilizes the text prompt as a *shared and universal encoding* of the various geometric controls (as shown in Fig. 9), which is more scalable, deployable, and computationally efficient.

**Advantage over methods with more complicated designs**   contains three perspectives, including: **1) Utilization of foundational pre-trained models.** Unlike GAN-based methods, GEODIFFUSION leverages large-scale pre-trained text-to-image diffusion models (*e.g.*, Stable Diffusion), enabling the generation of highly realistic and diverse detection data, which is crucial for data augmentation. **2) Transferability of the text encoder.** Different from existing methods (*e.g.*, GLIGEN (Li et al., 2023b)) requiring specifically designed bounding box encoding modules and training the parameters from scratch, GEODIFFUSION capitalizes on the transferability of text encoder (verified in Tab. 10, Rows 1 and 2), empowering more efficient adaptation and decreased need for the annotated training data, which is particularly beneficial for long-tailed classes with scarce data annotation. Specifically, GEODIFFUSION shows remarkable improvement in rare classes compared to GLIGEN, achieving **+11.9** and **+15.0** mAP for trailers and construction respectively, the Top-2 rare classes on NuImages, as reported in Tab. 2. **3) Usage of foreground prior re-weighting,** which significantly enhances the generation performance of foreground objects, as evident in Tab. 10 (Rows 2 and 5).

**Location translation via text.**   Although seeming cumbersome, the main purpose is to adopt **text as a universal encoding** for various geometric conditions and empower pre-trained T2I DMs for

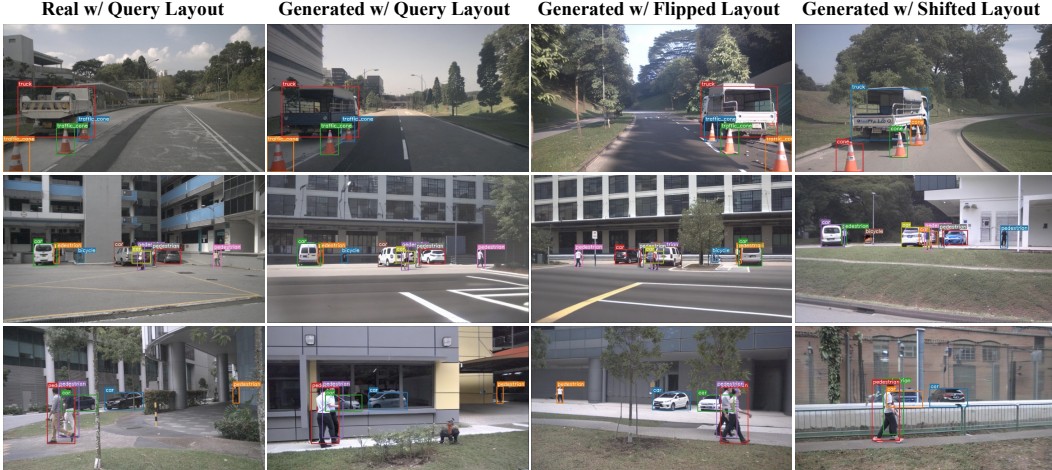

| Real w/ Query Layout | Generated w/ Query Layout | Generated w/ Flipped Layout | Generated w/ Shifted Layout |

Figure 6: **More visualization of generation generalizability for augmented bounding boxes**. GEODIFFUSION demonstrates superior performance for real-life collected and augmented layouts, consistently with what we have observed in Fig. 4.

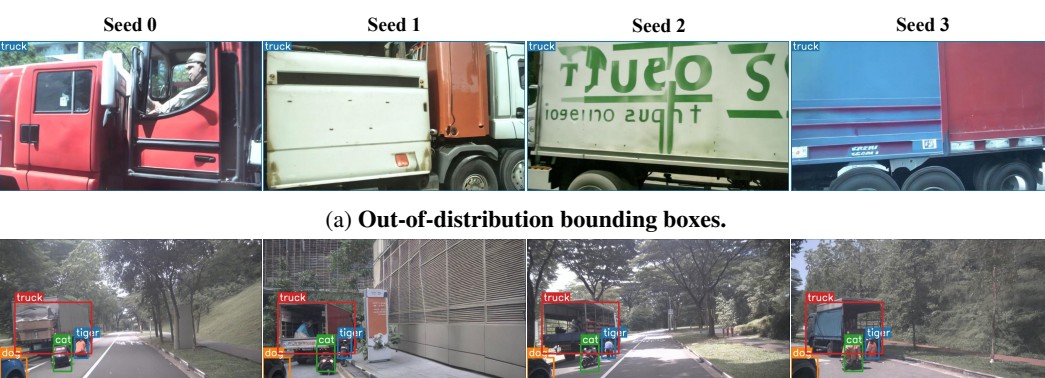

| Seed 0 | Seed 1 | Seed 2 | Seed 3 |

(a) **Out-of-distribution bounding boxes.**

(b) **Out-of-distribution classes.**

Figure 7: **More visualization of generation generalizability for totally out-of-distribution layouts.** Our GEODIFFUSION demonstrates strong robustness towards (a) OoD bounding boxes, but suffers from (b) unseen classes (*i.e.*, *dog, cat and tiger*) during fine-tuning.

detection data generation, which, however, might not support the dense pixel-level semantic control currently (*e.g.*, mask-to-image generation).

**Extendibility.** Thus, GEODIFFUSION can be extended to other descriptions as long as they can be discretized (*e.g.*, locations) or represented by text prompts (*e.g.*, car colors).

**Adaptation of the Foreground Re-weighting with existing methods** can be beneficial if applicable. However, existing methods utilize specific modules to encoder layouts (*e.g.*, RoI Align to take foreground features only in LAMA), suggesting that specific designs might still be required for the adaptation, which is beyond the scope of this work.

**Limitation.** We notice that the GEODIFFUSION-generated images by now can only contribute as augmented samples to train object detectors together with the real images. It is appealing to train detectors with generated images solely, and we will explore it in the future work. We hope that our simple yet effective method can bring researchers' attention to large-scale object detection *dataset* generation with more flexible and controllable generative models. The incorporation among GEODIFFUSION with the annotation generation (Reza et al., 2019; 2020) and even perception-based methods (Wu et al., 2023; Li et al., 2023c) is also appealing, which will be explored in the future.

Meanwhile, more flexible usage of the generated images beyond data augmentation, especially incorporation with generative pre-training (Chen et al., 2023a; Zhili et al., 2023), contrastive learning (Chen et al., 2021a; Liu et al., 2022b), is also an appealing future research direction.

# D  MORE APPLICATIONS

**Camera views**  are introduced in GEODIFFUSION primarily to demonstrate that text prompts can serve as a unified representation for various geometric conditions, facilitating independent manipulation without interdependencies. As illustrated in Fig. 9, simply converting the camera view tokens "{view}" can effectively generate images from different camera views while maintaining semantic consistency, revealing GEODIFFUSION's flexible ability to handle various geometric conditions.

**Domain adaptation**  among different weather conditions and times of day can be supported simply by integrating the extra conditions into the text prompts as, "A {weather} {time} image of {view} camera with {bboxes}". Fig. 10 showcases the capability of our GEODIFFUSION to flexibly adapt between daytime, rainy and night scenes.

**Long-tailed generation.**  We further train a GEODIFFUSION on the challenging LVIS (Gupta et al., 2019) dataset, an extremely long-tailed scenario, with the exact same optimization recipe with the COCO-Stuff as in Sec. 4.3, and provide a qualitative evaluation in Fig. 13, where the annotations of **"rare classes"** are highlighted in the images. As shown in Fig. 13, GEODIFFUSION demonstrates superior generation capabilities even for the long-tailed rare classes.

**3D geometric controls**  (*e.g.*, 3D locations, depth and angles) can be supported in GEODIFFUSION by projecting 3D bounding boxes into the 2D image planes. Specifically, the 3D LiDAR coordinates of all corners of the 3D bounding boxes are first projected into the 2D image plane as $\{(x_i, y_i)\}_{i=1}^{8}$, where $(x_i, y_i)$ denotes the projected $i$-th corner of the given 3D bounding box, and then discretized and encoded separately following the exact same manner with Sec. 3.2. Note that different from the 2D scenario, a 3D bounding box is determined by 8 corners. Thus, GEODIFFUSION can control the *3D locations* and *depth* with the same text-prompted geometric controls, as demonstrated in Fig. 11, while *angles* can be supported simply by **reversing** the encoding order of the 8 corners into the text prompts to change the object orientation, as shown in Fig. 11 (4th column). We will support more 3D geometric controls in the future work.

# E  MORE QUALITATIVE COMPARISON

We provide more qualitative comparison on NuImages, COCO-Stuff and LVIS datasets in Fig. 8-13.

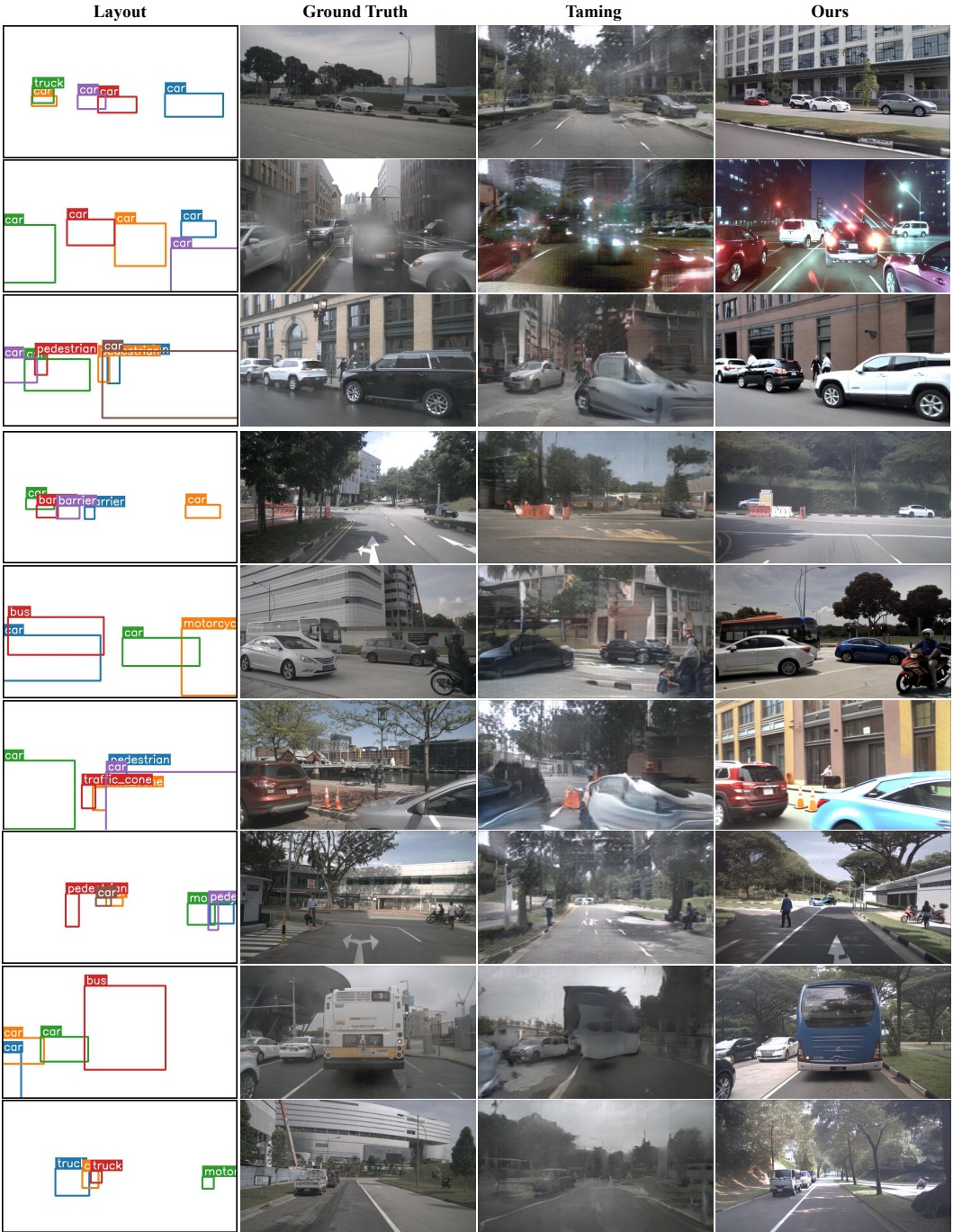

Figure 8: **More qualitative comparison on the NuImages (Caesar et al., 2020) dataset**.

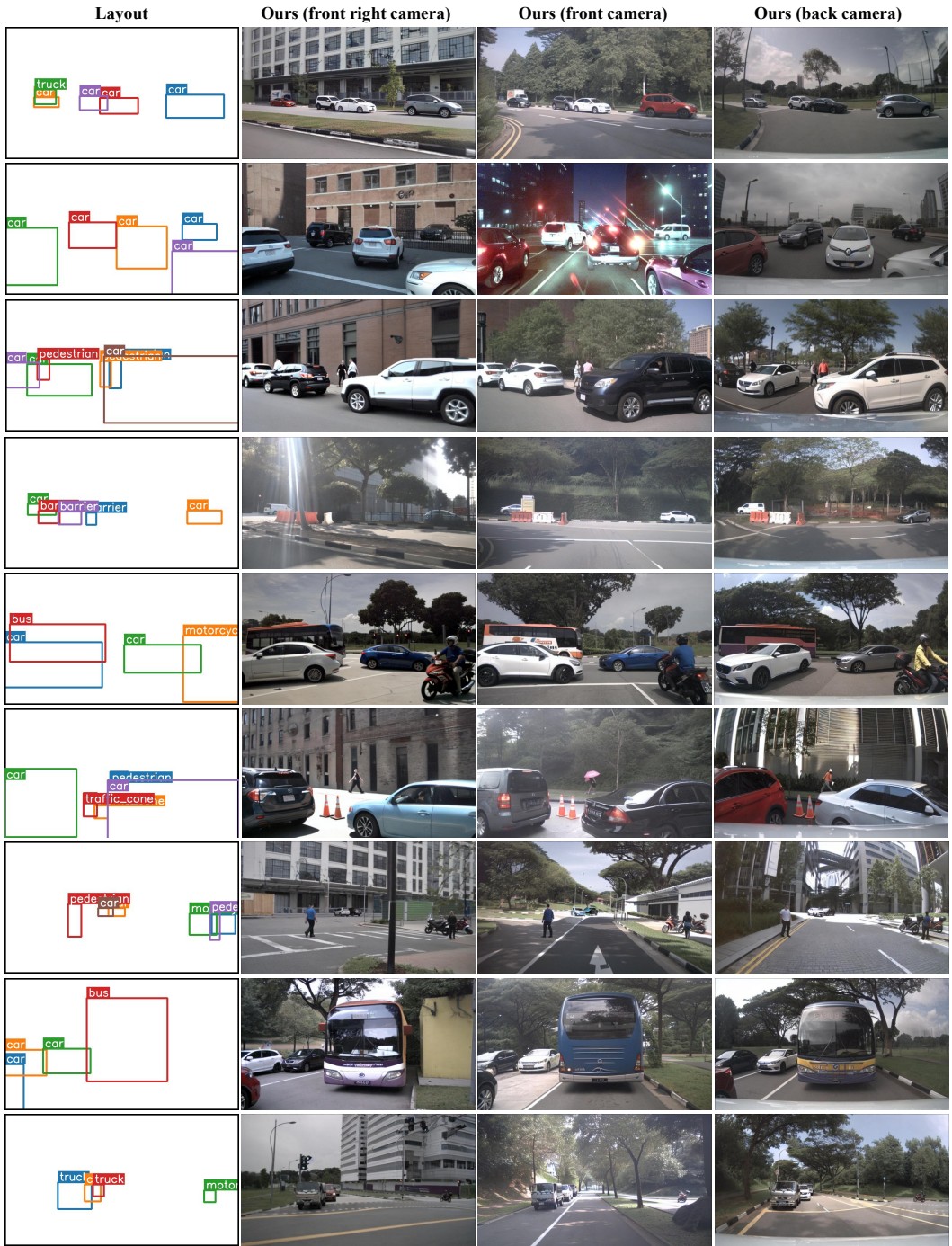

Figure 9: **More qualitative comparison for camera-dependent generation on NuImages (Caesar et al., 2020) dataset**.

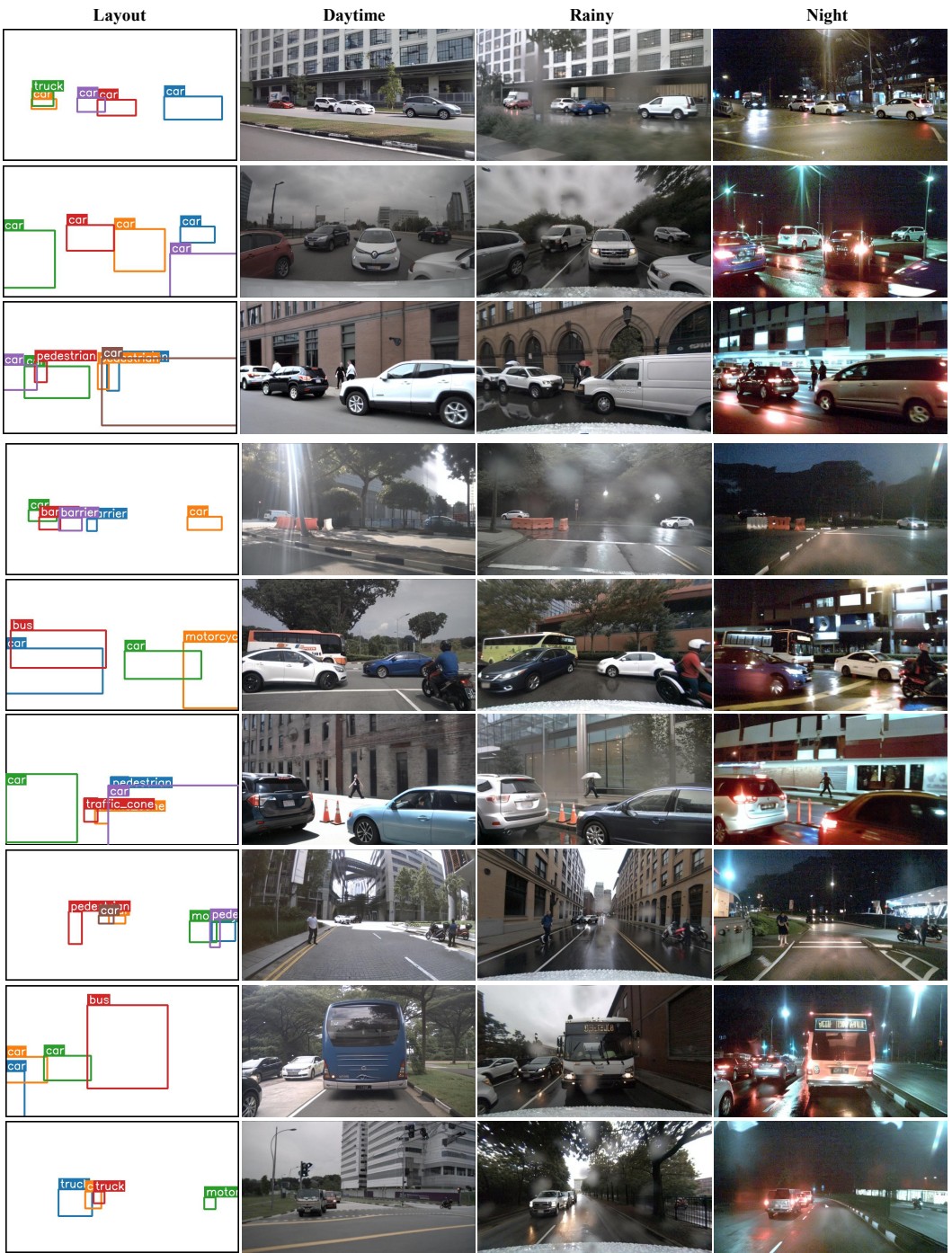

Figure 10: **More qualitative comparison for the weather and time day control generation on the NuImages (Caesar et al., 2020) dataset**.

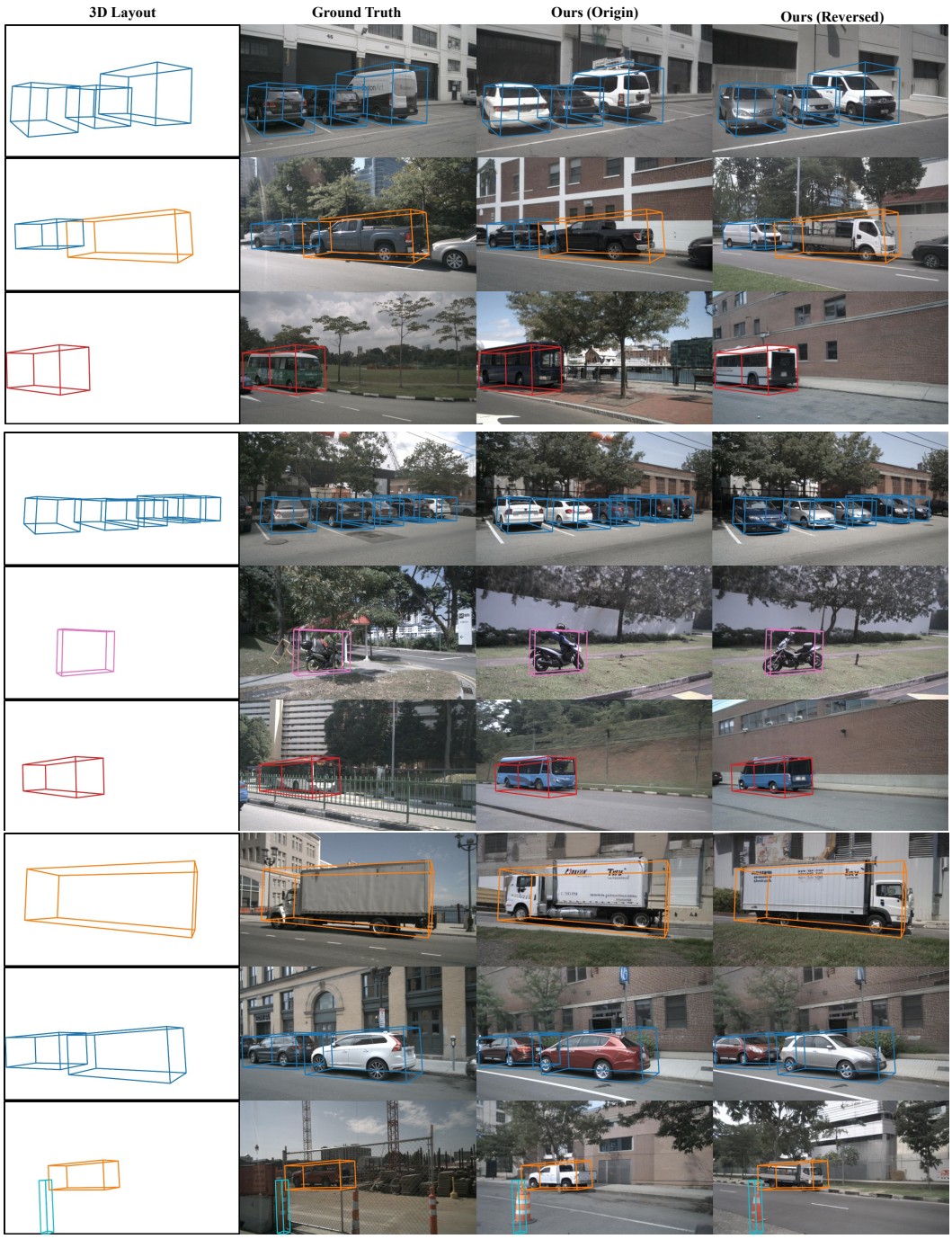

Figure 11: **More qualitative comparison for the 3D geometric controls on the NuScenes (Caesar et al., 2020) dataset**.

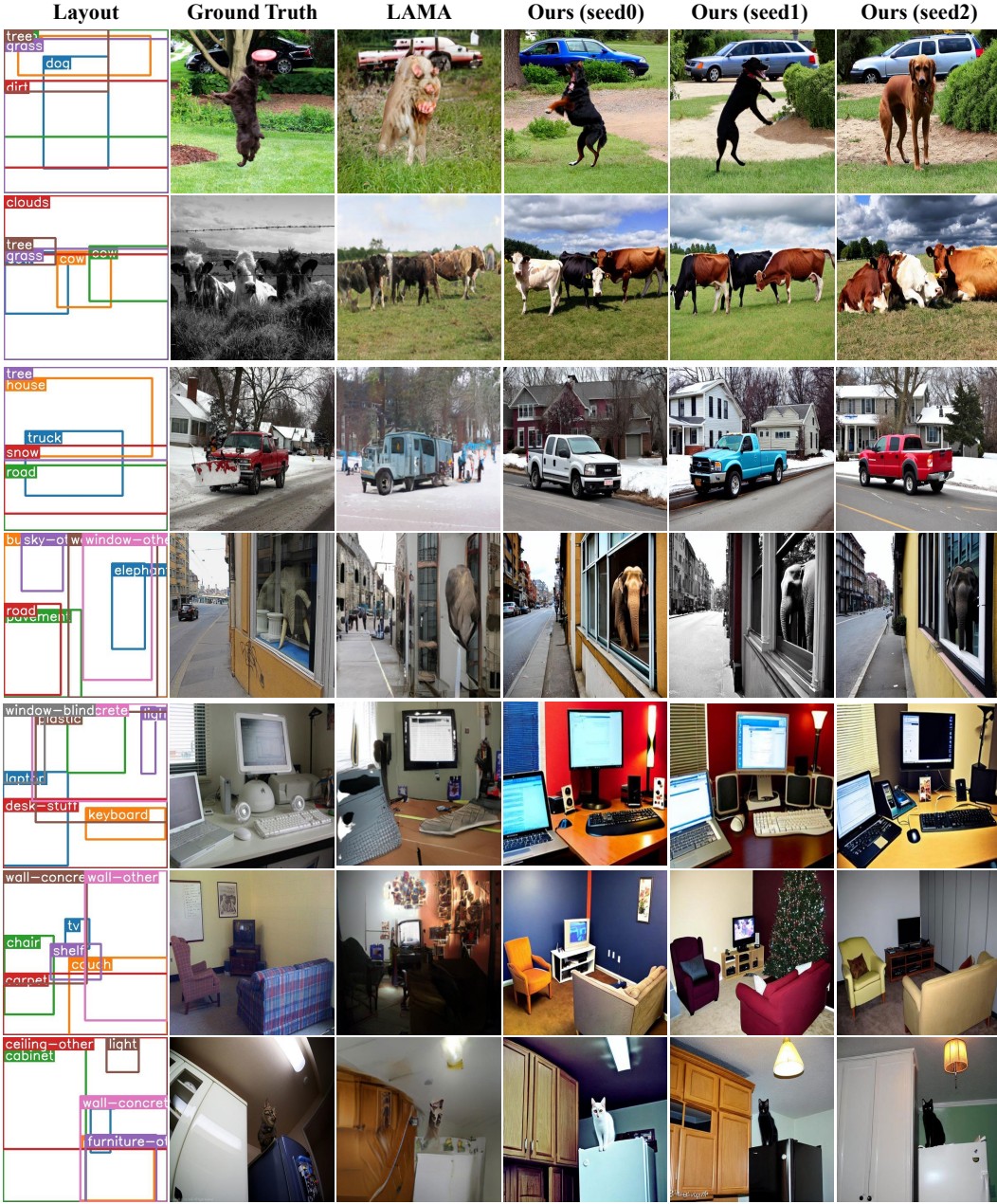

Figure 12: **More qualitative comparison on the COCO-Stuff (Caesar et al., 2018) dataset**. Our GEODIFFUSION can successfully deal with both outdoor (1st-4th rows) and indoor (5th-7th rows) scenes, while demonstrating significant fidelity and diversity (4th-6th columns are generated images by GEODIFFUSION under three different random seeds).

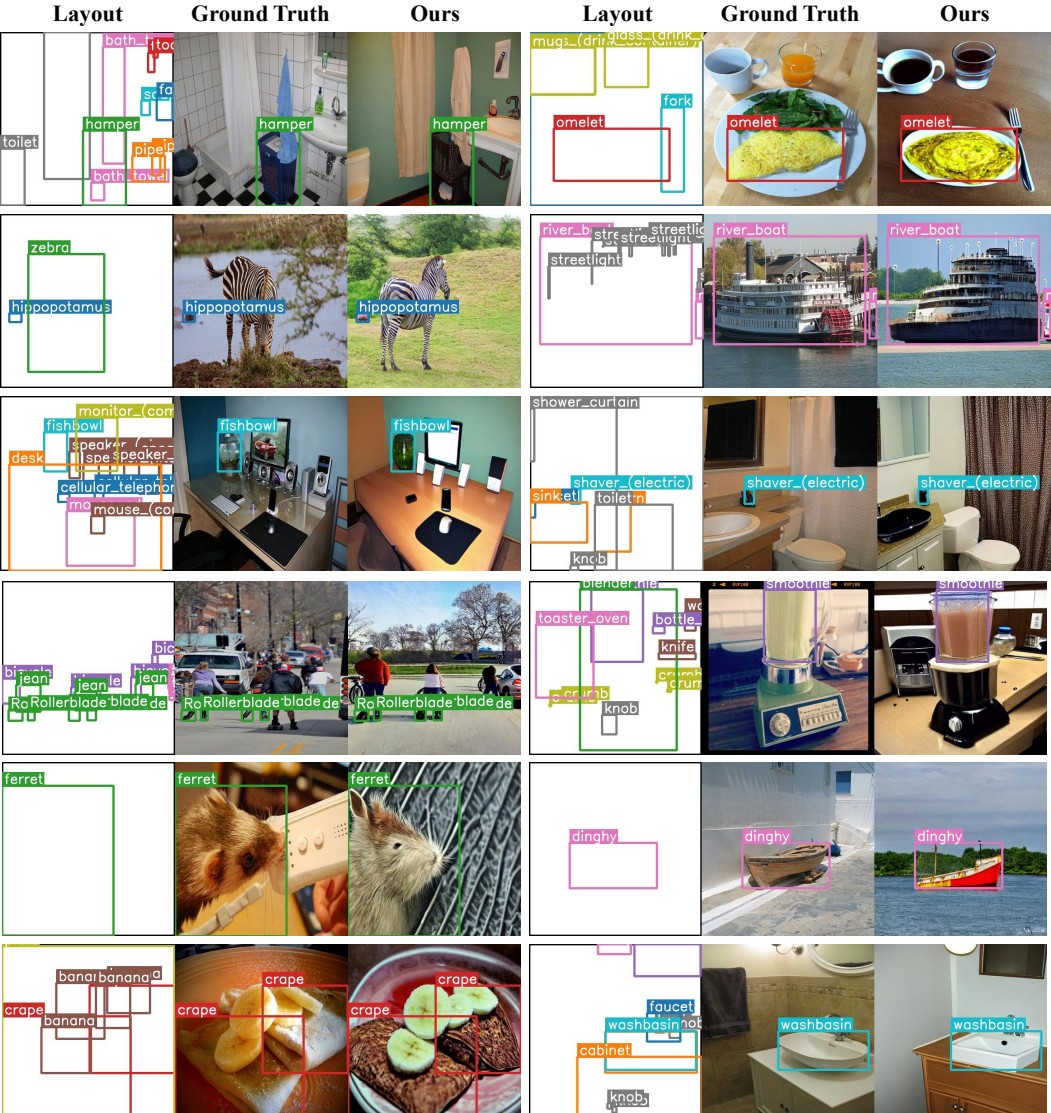

Figure 13: **More qualitative comparison on the LVIS (Gupta et al., 2019) dataset**. Within each group, we demonstrate the input layouts (left), the ground truth images (middle) and the generated images by GEODIFFUSION (right). We also highlight the annotations of long-tail rare classes on the images. Our GEODIFFUSION can successfully generate highly realistic images consistent with the given layouts, even for the long-tail rare classes.

