# OpenReview forum: "GeoDiffusion: Text-Prompted Geometric Control for Object Detection Data Generation"
_ICLR.cc/2024/Conference — ICLR 2024 poster_

### Official Review · Reviewer_VRFf · 2023-10-19

**Soundness:** 3 good
**Presentation:** 4 excellent
**Contribution:** 3 good
**Rating:** 6
**Confidence:** 4

**Summary:**

This paper aims to generate high-quality detection dataset via text-to-image diffusion models. The main novelty lies in GeoDiffusion is that it encodes not only the bounding boxes but also extra geometric conditions such as camera views in selfdriving scenes.
To embed the bounding box locations, it discretizes the continuous coordinates by dividing the image into a grid of location bins, creates unique location tokens, and inserts the  to the text encoder vocabulary. To embed other conditions along with bounding boxes, it uses a prompt template “An image of {view} camera with {boxes}”. To enforce the model focus on foreground and balance the bounding boxes of different sizes during training, it proposes an area re-weighting method to dynamically assign higher weights to smaller boxes.
It is claimed that GeoDiffusion outperforms previous L2I methods while maintaining faster training time.

**Strengths:**

The paper is well written. Despite the missing of some important baselines, the paper contains very thorough experiments, demonstrating the effectiveness of the proposed method.

**Weaknesses:**

1. The baselines, such as ControlNet and GLIGEN are missing in Table 3 and Table 6. Similarly ControlNet is missing in Table 5. They are apparently more comparable than those GAN-based methods. The claimed improvement in the introduction (+21.85 FID and +27.1 mAP) is questionable and misleading as they are obtained by comparing with old GAN-based methods.
2. Related to 1), Stable Diffusion is not a good option for comparison of layout-based inpainting in Figure 5 as Stable Diffusion does not consume bounding box explicitly. I suggest to replace it with ControlNet and GLIGEN for a fair comparison.
3. Related to 1), the importance of camera views is not clearly due to the missing baselines. Furthermore, there are no qualitative illustration of the effect of camera views except a single example in Figure 1.

**Questions:**

In Table 4, the proposed GeoDiffusion outperforms other methods except in AP50. ControlNet significantly outperform all the other method at AP50 by a very large margin. I am quite curious what leads to this? I think it worths some discussion in the paper.

---

> ### Author Response · Authors · 2023-11-21
> **Response to Reviewer VRFf**
>
> Thank you for your valuable and constructive reviews. Here we address your questions one by one.
>
> ---
>
> **W1. Additional baseline methods on NuImages and COCO-Stuff datasets.**
>
> Thank you for your suggestions. In the revision, we have 1) reported the fidelity and trainability performance of ControlNet and GLIGEN on the NuImages dataset in Table 2 and 3 respectively, 2) reported the trainability performance of ControlNet on the COCO dataset in Table 5, 3) reported the inpainting performance of ControlNet and GLIGEN on the COCO dataset in Table 6, and 4) refine the claimed improvement to be more specific in the introduction section. For detailed experimental results, please refer to **the common response to Reviewers WbtJ and VRFf**.
>
> ---
>
> **W2. Additional baseline methods for the inpainting comparison in Figure 5.**
>
> We provide a qualitative comparison between GeoDiffusion and Stable Diffusion in Figure 5 primarily to demonstrate how exactly the geometric-awareness enhances the inpainting ability. As shown in Figure 5, GeoDiffusion can better understand the requested inpainting locations especially under multi-object circumstances, and therefore, better deal with the inpainting requests.
>
> Following your suggestions, we further provide comparison with ControlNet and GLIGEN in Figure 5, where GeoDiffusion still demonstrates more superior inpainting results.
>
> ---
>
> **W3. More discussion on the camera view control generation.**
>
> **(1) Importance of camera views** has been verified by the ablation study conducted in Table 10 and Appendix B. Specifically, we train a GeoDiffusion without camera view tokens with all the other settings unchanged. As reported in Table 10 (rows 1 and 2), the variant without camera view tokens is surpassed by the default GeoDiffusion with respect to both the image quality (*i.e.*, FID) and layout consistency (*i.e.*, mAP), revealing the importance of camera views.
>
> **(2) More qualitative comparison of camera views** is provided in Figure 10 following your suggestions.
> As demonstrated, GeoDiffusion can effectively convert the camera views simply by changing the camera view tokens in text prompts, while maintaining consistent with the given semantic layouts.
>
> ---
>
> **Q1. Discussion on the high AP50 but low mAP and AP75 performance of ControlNet in Table 4.**
>
> This is primarily because ControlNet can only take image-level condition inputs instead of instance-level ones as GeoDiffusion.
>
> **(1) Implementation of ControlNet.** To facilitate the layout-to-image generation, ControlNet is implemented by converting the semantic layouts to $C$ binary semantic masks, with each mask for a single semantic class, following the ControlNet paper. Therefore, the model can no longer distinguish different instances of the same class. When two instances of the same class are close with each other, the model would tend to generate a single but large instance to cover the areas of both instances, which is a typical reason to obtain a high AP50 but low mAP and AP75.
>
> **(2) Qualitative verification.** A qualitative example is provided in Figure 5. As demonstrated, ControlNet completes the inpaiting requests by considering the left two people as a single instance, and therefore, only complements the right part of the human body.

---

> > ### Comment · Reviewer_VRFf · 2023-11-22
> > **Response to the authors**
> >
> > Thanks for the additional experiments and the clarifications. The response has solve the majority of my concerns. I have increased my rating.

---

> > > ### Author Response · Authors · 2023-11-23
> > > **Response to Reviewer VRFf**
> > >
> > > Thank you so much for your acknowledgement and positive evaluation of this work! We will keep improving and adding more geometric controls for GeoDiffusion!

---

### Official Review · Reviewer_qLFr · 2023-10-30

**Soundness:** 2 fair
**Presentation:** 3 good
**Contribution:** 3 good
**Rating:** 6
**Confidence:** 4

**Summary:**

This paper proposes GeoDiffusion that transforms locations/layouts of objects and view directions into a prompt which can be fed as input to text-to-image diffusion models. The generated data by the proposed GeoDiffusion is beneficial to improve object detection performance, especially for categories within the low-data regime.

**Strengths:**

The paper is written well. The method is simple and intuitive to understand.
The results are promising, especially in improving object detection performance.

**Weaknesses:**

Though the paper claims that the model supports flexible geometric conditions, the current model only supports location tokens and 6 view directions. However, geometric conditions may include depth, exact 3D locations and angles, etc. It is unclear whether the approach is “flexible” enough for these conditions.

It is unclear why the proposed method is better at trainability on object detection, compared to other layout-control synthesis methods such as GLIGEN. The authors mentioned that the method excels at low-data object categories but where does the advantage come from?

The paper exceeds the 9 page limit by ICLR, which violates the authors’ code of ICLR.

Minor typos. It should be “...embarrassingly simple…” rather than “...embarrassing simple…”

**Questions:**

See the second point in the weaknesses.

---

> ### Author Response · Authors · 2023-11-21
> **Response to Reviewer qLFr**
>
> Thank you for your acknowledgement and valuable suggestions. Here we address your questions one by one.
>
> ---
>
> **W1: Extension to 3D geometric controls.**
>
> Thank you for your insightful comments. It is nature for GeoDiffusion to intergrate various geometric controls, as long as they can be discretized and integrated into the text prompts. As highlighted, our current work primarily demonstrates the integration of camera views, a vital aspect in 2D object detection for autonomous driving, as evidenced in our ablation study (Table 10, rows 1 and 2).
>
> To address the extension into 3D domains, one needs to incorporate 3D conditions, such as depth, exact 3D locations, and angles, into the generation process. This can be achieved by projecting 3D LiDAR coordinates onto the 2D image plane, followed by text-prompted 2D conditional generation, akin to the process used in GeoDiffusion. Such methodologies have been preliminarily explored in recent studies [1, 2]. Limited by the scope of this paper, our primary focus is on data generation for 2D object detection. The implementation of support for 3D geometric controls is left as a subject for future work.
>
> ---
>
> **W2. Advantages of GeoDiffusion over methods such as GLIGEN, particularly in terms of trainability for object detection.**
>
> **(1) Utilization of foundational pre-trained models.** Unlike GAN-based methods, a significant advantage of GeoDiffusion lies in leveraging large-scale pre-trained text-to-image diffusion models, such as Stable Diffusion. These models enable the generation of highly realistic and diverse detection data, which is crucial for effective data augmentation.
>
> **(2) Transferability of the text encoder.** In contrast to existing methods like GLIGEN and ControlNet, which require designing specific bounding box encoding modules and training parameters from scratch, GeoDiffusion capitalizes on the transferability of the text encoder. This approach allows for more efficient adaptation and reduces the need for annotated training data. This is particularly beneficial for long-tailed classes with scarce data annotations. For instance, GeoDiffusion shows remarkable improvement in rare classes such as trailers and construction vehicles, achieving +11.9 and +15.0 mAP, respectively, compared to GLIGEN in the Top-2 rare classes on NuImages, as demonstrated in Table 2. The efficacy of the text encoder's transferability for geometric conditions is further supported by our ablation studies in Table 9 (rows 1 and 2).
>
> **(3) Usage of the foreground prior re-weighting.** This strategy significantly enhances generation performance by focusing on the modeling of foreground objects. The positive impact of this approach is evident in the comparison between the results in Table 9 (rows 2 and 5).
>
> ---
>
> **W3-W4. Paper formatting and typos.**
>
> Sorry for the careless mistakes. We have fixed the formatting and typo problems in the revision.
>
> ---
>
> [1] Gao R, Chen K, Xie E, et al. Magicdrive: Street view generation with diverse 3d geometry control[J]. arXiv preprint arXiv:2310.02601, 2023.
>
> [2] Yang K, Ma E, Peng J, et al. BEVControl: Accurately Controlling Street-view Elements with Multi-perspective Consistency via BEV Sketch Layout[J]. arXiv preprint arXiv:2308.01661, 2023.

---

> > ### Comment · Reviewer_qLFr · 2023-11-22
> > **After reading author response**
> >
> > Thanks for the authors to post responses to my questions. I'm satisfied with most but one remaining issue is designing a specific bounding box encoding module should not be counted as a drawback, as long as it works well. Besides, the proposed method also leverages annotation of bounding boxes during training, which makes no huge difference to those approaches. I also enjoy a more free-form of controllability. It is just that the comparison, especially the rationale, to GLIGEN style layout-to-image approaches should be made more clear and sound.

---

> > > ### Author Response · Authors · 2023-11-23
> > > **Response to Reviewer qLFr**
> > >
> > > Thank you so much for your valuable responses. Here we address your concerns one by one.
> > >
> > > ---
> > >
> > > **Q1: Further comparison with existing layout-to-image approaches.**
> > >
> > > **(1) Encoding of bounding boxes.** Although achieving reasonable performance, existing layout-to-image approaches utilizing specifically designed bounding box encoding modules like GLIGEN are still 1) significantly surpassed by our GeoDiffusion with the simplest text-prompted controls on both the NuImages and COCO-Stuff datasets and 2) not flexible for other geometric conditions like camera views, suggesting that the delicate design of bounding box encoding is possible but evitable.
> > >
> > > **(2) Usage of bounding box annotation.** Indeed, GeoDiffusion still requires bounding box annotation data to learn geometric controls. However, with the same training dataset, GeoDiffusion still demonstrate non-trivial improvement upon all baseline methods, revealing the superiority of our proposed method. We will explore how to introduce geometric controls without data with bounding box annotation in the future work.

---

### Official Review · Reviewer_wbtJ · 2023-10-30

**Soundness:** 3 good
**Presentation:** 2 fair
**Contribution:** 2 fair
**Rating:** 6
**Confidence:** 4

**Summary:**

This paper proposes a method that utilizes a pretrained T2I diffusion model, e.g., Stable Diffusion, for generating images given the box layout. Also, the pair of images and provided box layouts can be used to train an object detector. They propose a technique to encode the box layout into the text prompt and fine-tune the whole network except the VQ-VAE. Furthermore, the authors propose a reweighting mask for balancing between foreground and background regions to generate small objects with higher quality. The experiments show the effectiveness of their methods in three aspects: fidelity, trainability, and generability.

**Strengths:**

1. The ablation study is done thoroughly including the parameter choices.
2. This paper provides a clear metric for validating their results against other generative methods. For fidelity, trainability, and universality, this paper surpasses other methods by a large margin.
3. The authors show the effectiveness of this method in the aspect of data augmentation for training an object detector.

**Weaknesses:**

1. Given the fact that recent diffusion-based methods such as ControlNet, ReCo, and GLIGEN can also do Layout-to-Image tasks, the authors should include them in Table 2 (Comparison of generation fidelity on NuImages) and Table 3 (Comparison of generation trainability on NuImages). Current comparisons in Tables 2 and 3 are not up-to-date.
2. The proposed camera-dependent conditioning prompt is very similar to the view-dependent prompting in DreamFusion. The authors need to ablate more on why their method is novel compared to the one proposed in DreamFusion.
3. The idea of using synthetic images generated from the model to augment training object detectors is reasonable. However, the authors need to ablate how much data is needed to train GeoDiffusion (in the Trainability section), since if the number of real data (and box annotations) required to train GeoDiffusion is significantly larger than the numbers needed to train an object detector, the application of this paper is not realistic.
4.  The method is sensitive to the image size and aspect ratio of each dataset. In other words, the number of bins is different across datasets, therefore, greatly affecting the transferability of the proposed approach when trained on a dataset and tested on another dataset. In contrast, other baselines such as Copy and Paste Synthesis can use Stable Diffusion to generate the object of interest better in this situation.
5. Lack of experiments of long-tail (rare) object detection datasets such as LVIS and domain adaptation such as GTAV to Cityscapes or Cityscapes to Foggy Cityscapes. In these settings, the importance of synthetic data is more significant than in the domain where real images and box annotations are largely available.
6. Among these two contributions, which one is more important?
7. It would be better to move the ablation study from supp (table 9, 10) to the main paper.

**Questions:**

1. Does the proposed camera-view conditioning work on other autonomous-driving datasets such as Waymo Open Dataset
2. Can the model generate rare cases in the dataset? For example, can the model generate diverse night-time, foggy, raining scenes in NuImages? Another interesting question is can the model generate more samples of rare classes on a long-tailed dataset (LVIS).
3. In Table 2 (Comparison of generation fidelity on NuImages), there is a version of GeoDiffusion with an input resolution of 800x456, how is this obtainable while Stable Diffusion input is 512x512?
4. How do the location bins resolutions affect generation in testing? For instance, if the location bins are trained at 256x256, can the model generate a prompt with a 512x512 grid in test time?

---

> ### Author Response · Authors · 2023-11-21
> **Response to Reviewer wbtJ (1/2)**
>
> Thank you for your valuable and constructive reviews. Here we address your questions one by one.
>
> ---
>
> **W1. Additional baseline methods on NuImages in Tables 2 and 3.**
>
> As discussed in Section 4.3, we compared GeoDiffusion with recent diffusion-based methods, including ControlNet, ReCo, and GLIGEN, on the COCO-Stuff dataset using publicly available checkpoints. In these comparisons, GeoDiffusion has demonstrated significant superiority.
>
> Following your suggestions, we have also reported the performance of ControlNet, ReCo, and GLIGEN on the NuImages dataset in Tables 2 and 3, where GeoDiffusion continues to show consistent improvement.
> For detailed experimental results, please refer to **the common response to Reviewers WbtJ and VRFf**.
>
> ---
>
> **W2. Comparison between the camera-dependent generation in GeoDiffusion and the view-dependent prompting in DreamFusion.**
>
> **(1) Controllability.** In GeoDiffusion, camera view tokens are explicitly utilized as input conditions to generate images from different camera views. This approach differs fundamentally from DreamFusion, where view-dependent prompting serves primarily as prior knowledge to enhance the supervision signal in training the NeRF model. DreamFusion still relies on explicit camera parameters to generate images from various views. In contrast, our method leverages direct input conditions, offering more intuitive controllability over the image generation process.
>
> **(2) Purpose.** Our primary goal with camera views in GeoDiffusion is not to introduce a novel view encoding method but to demonstrate that text prompts can serve as a unified representation for various geometric conditions, facilitating independent manipulation without interdependencies. As illustrated in Figure 10, the simple conversion of camera view tokens in GeoDiffusion can directly generate images from different perspectives while maintaining semantic integrity. This showcases our method's ability to handle geometric conditions more flexibly than DreamFusion.
>
> ---
>
> **W3. Ablation study on the amount of real data required to effectively train GeoDiffusion.**
>
> In Section 4.2.2, under the "necessity of real data" paragraph, we detail an experiment designed to ascertain the minimum quantity of images needed to train GeoDiffusion efficiently.
> Our approach involves the "subset" mode, where we randomly select subsets of the real image dataset (*e.g.*, 10%, 25%, 50%, and 75%) for training individual GeoDiffusion models. Each of these models then generates synthetic images to augment their respective subsets. Subsequently, these augmented datasets, comprising both real and synthetic images, are used to train object detectors.
>
> As illustrated in Figure 1(b) and reported in Table 8, our findings reveal that GeoDiffusion can achieve consistent performance improvements even with as little as 10% of the real images. Notably, the benefits of GeoDiffusion are more pronounced under data-scarce conditions. This demonstrates that the amount of real data required to train GeoDiffusion is considerably less than what would typically be needed to train an object detector directly, thereby enhancing the practicality and applicability of our approach.
>
> ---
>
> **W4. Transferability across datasets and comparison with copy-and-paste synthesis.**
>
> **(1) Different number of bins across datasets affecting transferability.** To address your concern, we would like to clarify that in our experiments, all optimization hyperparameters, including the number of pixels in each location bin, learning rates, and foreground loss weights, have been consistently maintained across both datasets (NuImages and COCO-Stuff). We acknowledge that we have not explicitly tested the model's transferability from one dataset to another. However, the diversity of classes and scenarios covered in NuImages and COCO-Stuff provide a robust assessment of our method's transferability. The distinct scenarios presented in the training and test sets of NuImages, in particular, attest to the effectiveness of our approach across varied contexts.
>
> **(2) Comparison with copy-and-paste synthesis.** In the comparison with copy-and-paste methods, it's important to highlight that our paper's primary objective is to introduce a unified framework capable of not only generating high-quality detection data but also supporting geometric-aware image generation and inpainting requests. While copy-and-paste methods may offer greater flexibility in data augmentation, they typically fall short in generating realistic images and providing additional geometric controls. For a more comprehensive comparison with related works, we invite you to review the detailed analyses presented in Table 1 and Section 2 of our main paper.

---

> > ### Author Response · Authors · 2023-11-21
> > **Response to Reviewer wbtJ (2/2)**
> >
> > **W5-1 & Q2. Support of GeoDiffusion for long-tailed object detection.**
> >
> > **(1) Long-tailed generation on NuImages (semantic classes).** The NuImages dataset is inherently long-tailed, with rare classes such as trailers and construction vehicles comprising only 1.5% of training annotations (Page 7, second paragraph). Tables 2 and 3 demonstrate GeoDiffusion's efficacy in addressing the long-tailed object detection challenges.
> >
> > **(2) Long-tailed generation on NuImages (weathers and time of day).** GeoDiffusion can generate images under different weather conditions and times of day through tailored text prompts (*e.g.*, **"A {weather} {time} image of {camera} camera with {bboxes}"**). Figure 9 showcases this ability, though limitations exist, such as the absence of foggy images in the NuImages dataset.
> >
> > **(3) Application to the LVIS Dataset.** Thank you for your suggestions. The LVIS dataset, representing an extremely long-tailed scenario, is still a nightmare even for state-of-the-art models [1]. This suggests that adapting GeoDiffusion to LVIS requires a specifically designed optimization recipe, which we plan to explore in the future work.
> >
> > ---
> >
> > **W5-2. Support of GeoDiffusion for domain adaptation.**
> >
> > GeoDiffusion shows adaptability to various domains like daytime, rainy, and night scenes, as shown in Figure 9. We aim to extend this capability to more diverse domain adaptation scenarios in our future work.
> >
> > ---
> >
> > **W6. Prioritization of contributions for long-tailed detection and domain adaptation.**
> >
> > Both long-tailed detection and domain adaptation are crucial for real-life applications. Currently, we prioritize long-tailed detection due to its immediate feasibility and impact. However, we are committed to further exploring novel applications, including domain adaptation, for GeoDiffusion.
> >
> > ---
> >
> > **W7. Suggestion on ablation study placement.**
> >
> > Thank you for your suggestion to include Tables 9 and 10 in the main paper. Due to page limitations, we cannot move these tables directly. However, we have made clear references in Section 4.4 to these tables in the supplementary material, ensuring easy accessibility for readers interested in our detailed ablation studies.
> >
> > ---
> >
> > **Q1. Discussion on the camera-view conditional generation.**
> >
> > The camera-view conditioning works as long as the camera view information is provided and integrated into the text prompts. In this paper, we mainly conduct experiments on the NuImages dataset, a typical autonomous driving dataset featuring multiple diverse driving views.
> > Given a larger and more diverse dataset like the Waymo Open dataset, we anticipate that GeoDiffusion would exhibit even more enhanced camera view control capability.
> >
> > ---
> >
> > **Q3. Support for downstream fine-tuning resolution besides 512 $\times$ 512.**
> >
> > The UNet of the Stable Diffusion mainly consists of convolutional blocks and Transformer blocks, both of which are invariant to the input resolution. Therefore, we can flexibly convert the image resolution during fine-tuning.
> >
> > ---
> >
> > **Q4. Effect of the grid size of location bins during testing time.**
> >
> > To clarify, the grid size of location bins should be consistent between the training and testing time, otherwise the model can not correctly understand the given bounding boxes, and therefore, fail to generate objects in the specified locations.
> >
> > ---
> >
> > [1] Zhang Y, Kang B, Hooi B, et al. Deep long-tailed learning: A survey[J]. IEEE Transactions on Pattern Analysis and Machine Intelligence, 2023.

---

> ### Comment · Reviewer_wbtJ · 2023-11-23
>
> Thanks the authors for their response. I found the response just partly addressed my concerns. The concern regarding training in one dataset and testing in another dataset (different image sizes or aspect ratios in transfer testing) is not addressed, so basically, cannot be handled right? Also, the long-tail and domain adaptation is not addressed either. Therefore, I can only marginally increase my score to 6.

---

> ### Author Response · Authors · 2023-11-23
> **Response to Reviewer wbtJ**
>
> Thank you so much for your acknowledgement of this work. Here we address your concerns one by one.
>
> ---
>
> **Q5. Transfer between datasets with different image sizes and aspect ratios.**
>
> Usually, it is difficult to require large-scale image datasets like COCO-Stuff to have a unified image size and aspect ratio for all the images.Therefore, in order to perform mini-batch-based optimization, we follow common practices [2, 3, 4] to first resize all images to a unified resolution $H\times W$ (*e.g.*, $H\times W=512\times512$ for the default Stable Diffusion) as the model input shape during training, while during testing, the model will first generate images of $H\times W$, which are then resized to the original image resolution through (bilinear) interpolation. Thus, transfer between different datasets with different image sizes and asepct ratios would not be a problem here.
>
> ---
>
> **Q6. Further discussion on the application for long-tailed and domain adaptation.**
>
> **(1) Further discussion on long-tailed generation.** Following your suggestions, we conduct a preliminary experiment on the LVIS dataset with the exact same optimization recipe with the COCO-Stuff dataset as discussed in Section 4.3, and provide a qualitative evaluation in Figure 12, where the annotations of **"rare classes"** are highlighted in the images. As shown in Figure 12, GeoDiffusion demonstrates superior generation capabilities even for the long-tailed rare classes. We will keep improving the performance in the final revision.
>
> **(2) Further discussion on domain adaptation.** As shown in Figure 9, GeoDiffusion has already demonstrated flexible domain adaptation capabilities between **daytime, rainy and night** circumstances, while the generation of foggy images are currently not support primarily due to the **absence of foggy images** in the NuImages dataset. We will further explore this specific application in the future work.
>
> *Besides, we notice that the score of the paper in the OpenReview System has not been changed. To complete the score-changing procedure, you might also need to edit the **"Rating"** option in your official review. Thank you.*
>
> ---
>
> [2] Rombach R, Blattmann A, Lorenz D, et al. High-resolution image synthesis with latent diffusion models. In CVPR. 2022.
>
> [3] https://github.com/CompVis/latent-diffusion
>
> [4] https://github.com/huggingface/diffusers/

---

### Official Review · Reviewer_2g8v · 2023-10-31

**Soundness:** 3 good
**Presentation:** 4 excellent
**Contribution:** 3 good
**Rating:** 8
**Confidence:** 4

**Summary:**

The paper proposes a data generation pipeline to utilize diffusion models to generate text-prompted data with flexible geometric controls. The results are impressive in many 2D applications. It proves that the generated data can be used to improve the training of object detectors, providing the effectiveness of this line of research.

**Strengths:**

+ The paper is easy to follow with clear motivation. The writing is good.

+ The experiments are extensive with clear explanations. It is verified on a wide variety of tasks.

+ It is verified that the layout-image models could be used to facilitate the conventional object detection pipeline, which is very useful.

**Weaknesses:**

Some results need to be discussed in more details. How to apply into the 3D domain deserves to be discussed. Please see questions below.

**Questions:**

1. Inconsistent performance in some metrics. For example, FID with 512 setting is better in Table 2. The mAP metric for Ped and Cone is slightly lower in Table 3. Any discussions on this?

2. As claimed in Section 4.2.1, the proposed method achieves 4X acceleration in the training procedure, how is it measured?

3. The pipeline is quite straightforward with diffusion process in an encoder-decoder architecture. The results are impressive. What's the key ingredient? Why previous methods with more complicated design fail to achieve the performance?

---
Minor: catpion in Figure 2 (b), should be "utilized" to train.

---

> ### Author Response · Authors · 2023-11-21
> **Response to Reviewer 2g8v (1/2)**
>
> Thank you for your acknowledgement and valuable suggestions. Here we address your questions one by one.
>
> ---
>
> **W1: Extension to 3D Geometric Controls.**
>
> Thank you for your insightful comments. It is nature for GeoDiffusion to integrate various geometric controls, as long as they can be discretized and integrated into the text prompts. As highlighted, our current work primarily demonstrates the integration of camera views, a vital aspect in 2D object detection for autonomous driving, as evidenced in our ablation study (Table 10, rows 1 and 2).
>
> To address the extension into 3D domains, one needs to incorporate 3D conditions, such as depth, exact 3D locations, and angles, into the generation process. This can be achieved by projecting 3D LiDAR coordinates onto the 2D image plane, followed by text-prompted 2D conditional generation, akin to the process used in GeoDiffusion. Such methodologies have been preliminarily explored in recent studies [1, 2]. Limited by the scope of this paper, our primary focus is on data generation for 2D object detection. The implementation of support for 3D geometric controls is left as a subject for future work.
>
> ---
>
> **Q1. Justification of inconsistent performance metrics.**
>
> **(1) FID score for the 512 $\times$ 512 model is better than that of the 800 $\times$ 456 variant in Table 2.** This can be attributed to the alignment between the pre-trained resolution of the Stable Diffusion model and the fine-tuning resolution of the 512 $\times$ 512 model. The Stable Diffusion model, in our study, is originally pre-trained at a resolution of 512 $\times$ 512. When we fine-tune this model at the same resolution (512 $\times$ 512), it results in a more effective transfer of learned features and parameters. This alignment of resolutions minimizes the discrepancies that might arise from adapting to different resolutions, and thereby enhancing the model's performance in terms of the FID score. However, when fine-tuning at a different resolution (as in the case of the 800 $\times$ 456 variant), the model faces the challenge of adapting to a new resolution, which can lead to a slight decrease in FID score.
>
> **(2) Lower mAP for pedestrian and traffic cone than the real-image-only baseline in Table 3.** In Section 4.2.1, we discuss the challenges that GeoDiffusion faces, particularly with the high variance and small objects, such as pedestrians and traffic cones, where the performance of our model is slightly lower compared to the Oracle real-image-only baseline. This indicates that the quality of the generated data for these two categories may not be sufficient to significantly enhance the performance of object detectors. Similar difficulties are encountered by other generative models as well. As shown in Table 3, except for the real-image baseline, our method outperforms other generative competitors, including LostGAN, LAMA, Taming, ReCo, GLIGEN, and ControNet. We are the first to demonstrate that generated data can serve as effective data augmentation to improve the overall performance of object detectors. We will continue to focus on improving the unsatisfactory performance in generating high variance and small objects in our future work.
>
> ---
>
> **Q2. Clarification on the 4X training acceleration in Section 4.2.1.**
>
> Sorry for the confusion caused. The measure of training acceleration is calculated based on the number of training epochs required to achieve a comparable level of performance. In Table 2, we demonstrate that our GeoDiffusion model, trained for only 64 epochs, significantly outperforms models like LostGAN, LAMA, and Taming, which were trained for 256 epochs. This indicates that GeoDiffusion achieves the same or better performance four times faster in terms of epochs, hence the 4X acceleration in training. To ensure clarity, we have revised the manuscript to explain this comparison more explicitly.

---

> > ### Author Response · Authors · 2023-11-21
> > **Response to Reviewer 2g8v (2/2)**
> >
> > **Q3. Key factors contributing to the impressive performance of GeoDiffusion.**
> >
> > **(1) Utilization of foundational pre-trained models.** Unlike GAN-based methods, a significant advantage of GeoDiffusion lies in leveraging large-scale pre-trained text-to-image diffusion models, such as Stable Diffusion. These models enable the generation of highly realistic and diverse detection data, which is crucial for effective data augmentation.
> >
> > **(2) Transferability of the text encoder.** In contrast to existing methods like GLIGEN and ControlNet, which require designing specific bounding box encoding modules and training parameters from scratch, GeoDiffusion capitalizes on the transferability of the text encoder. This approach allows for more efficient adaptation and reduces the need for annotated training data. This is particularly beneficial for long-tailed classes with scarce data annotations. For instance, GeoDiffusion shows remarkable improvement in rare classes such as trailers and construction vehicles, achieving +11.9 and +15.0 mAP, respectively, compared to GLIGEN in the Top-2 rare classes on NuImages, as demonstrated in Table 2. The efficacy of the text encoder's transferability for geometric conditions is further supported by our ablation studies in Table 9 (rows 1 and 2).
> >
> > **(3) Usage of the foreground prior re-weighting.** This strategy significantly enhances generation performance by focusing on the modeling of foreground objects. The positive impact of this approach is evident in the comparison between the results in Table 9 (rows 2 and 5).
> >
> > ---
> >
> > **Q4: Wording and typos.**
> >
> > Thanks. We have fixed the typos in the revision.
> >
> > ---
> >
> > [1] Gao R, Chen K, Xie E, et al. Magicdrive: Street view generation with diverse 3d geometry control[J]. arXiv preprint arXiv:2310.02601, 2023.
> >
> > [2] Yang K, Ma E, Peng J, et al. BEVControl: Accurately Controlling Street-view Elements with Multi-perspective Consistency via BEV Sketch Layout[J]. arXiv preprint arXiv:2308.01661, 2023.

---

> ### Comment · Reviewer_2g8v · 2023-11-22
>
> Thanks for the clarification! I have read the rebuttal and most of my concerns have been addressed. I would raise my score to clear accept.

---

> > ### Author Response · Authors · 2023-11-22
> > **Response to Reviewer 2g8v**
> >
> > Thank you so much for your positive evaluation of this work. It is really encouraging! We will persist in exploring the extension to 3D geometric controls for GeoDiffusion, making it a focal point for our future research directions.

---

### Author Response · Authors · 2023-11-21
**To All Reviewers**

We thank all reviewers for their time, insightful suggestions, and valuable comments. We are glad that ALL reviewers acknowledge the **"benefit of our work for object detection with throughout experiments"**, and find our work with **"good paper writing"** (Reviewer 2g8v, qLFr and VRFf) and **"surpassing other methods by a large margin"** (Reviewer wbtJ).

Below, we respond to each reviewer’s comments in detail. We have also revised the main paper and the appendix according to the reviewers’ suggestions. The main changes are listed as follows:

**In the main paper:**

1. Fidelity comparison on NuImages for ReCo, GLIGEN and ControlNet in Table 2.
2. Trainability comparison on NuImages for ReCo, GLIGEN and ControlNet in Table 3.
3. Trainability comparison on COCO for ControlNet in Table 5.
4. Inpainting quantitative comparison on COCO for GLIGEN and ControlNet in Table 6.
5. Inpainting qualitative comparison on COCO for GLIGEN and ControlNet in Figure 5.
6. Revise inappropriate wording and typos for better readability.

**In the appendix:**

1. Qualitative comparison for the weather and time of day control on NuImages in Figure 9.
2. Qualitative comparison for the camera view control on NuImages in Figure 10.
3. Qualitative evaluation on LVIS in Figure 12.

Note that we have marked all the revision in blue. We hope that our efforts can address the reviewers’ concerns well. Thank you very much again!

Best regards,

Paper 145 Authors

---

### Author Response · Authors · 2023-11-21
**Common Response to Reviewers wbtJ and VRFf: Additional Experiments**

**Reviewer wbtJ, Weakness 1 & Reviewer VRFf, Weakness 1 - Additional baseline methods on NuImages and COCO-Stuff datasets.**

Thank you for your suggestions. We conduct additional experiments on the mentioned baselines, including ReCo, GLIGEN and ControNet. The results are provided in Tables R1-4. We also revise Tables 2-6 in the main paper accordingly.

**(1) Fidelity performance of ReCo, ControlNet and GLIGEN on the NuImages dataset in Table 2.** As shown in Table R1, GeoDiffusion demonstrates superior fidelity performance compared to baseline methods such as ReCo, ControlNet and GLIGEN on the NuImages dataset. This is evidenced by achieving the lowest FID value of 9.58 and highest mAP value of 31.8, which is significantly better than the other methods listed. The results highlight the advanced capability of GeoDiffusion in generating high-quality detection data.

**(2) Trainability performance of ReCo, ControlNet and GLIGEN on the NuImages dataset in Table 3.** Table R2 highlights the superior trainability performance of GeoDiffusion compared to baseline methods like ReCo, ControlNet and GLIGEN on the NuImages dataset. GeoDiffusion achieves a notable improvement in mAP, with a score of 38.3. This is a significant increase of 1.9 mAP over ControlNet. The improvement is consistent across all categories, such as cars, trucks, and trailers, further emphasizing the enhanced trainability of our method.

**(3) Trainability performance of ControlNet on the COCO dataset in Table 5.** Table R3 demonstrates the enhanced trainability performance of GeoDiffusion compared to baseline methods such as ControlNet and GLIGEN on the COCO dataset. GeoDiffusion shows a significant improvement in mAP, achieving a score of 38.4. This represents a substantial increase of 1.5 mAP points over ControlNet.

**(4) Inpainting performance of ControlNet and GLIGEN on the COCO dataset in Table 6.**
Table R4 showcases the superior inpainting performance of GeoDiffusion compared to ControlNet and GLIGEN on the COCO dataset. GeoDiffusion leads with a mAP of 19.0, surpassing ControlNet's 17.8 and GLIGEN's 18.3. Additionally, GeoDiffusion achieves the highest scores in AP50 and AP75, further confirming its effectiveness in COCO dataset inpainting tasks.

▾ Table R1. **Comparison of generation fidelity on NuImages.**

| Method           | Resolution | Epoch | FID      | mAP      | AP50 | AP75 | APm | APl | trailer  | const.   | ped.     | car      |
| :--------------: | :--------: | :---: | :------: | :------: | :-------: | :-------: | :------: | :------: | :------: | :------: | :------: | :------: |
| ReCo             | 512 $\times$ 512    | 64    | 27.10    | 17.1     | 41.1      | 11.8      | 10.9     | 36.2     | 8.0      | 11.8     | 7.6      | 31.8     |
| GLIGEN           | 512 $\times$ 512    | 64    | 16.68    | 21.3     | 42.1      | 19.1      | 15.9     | 40.8     | 8.5      | 14.3     | 14.7     | 38.7     |
| ControlNet       | 512 $\times$ 512    | 64    | 23.26    | 22.6     | 43.9      | 20.7      | 17.3     | 41.9     | 10.5     | 15.1     | 16.7     | 40.7     |
| **GeoDiffusion** | 512 $\times$ 512    | 64    | **9.58** | **31.8** | **62.9**  | **28.7**  | **27.0** | **53.8** | **21.2** | **27.8** | **18.2** | **46.0** |

▾ Table R2. **Comparison of generation trainability on NuImages.**

| Method           | mAP      | car      | truck    | trailer  | bus      | const.   | bicycle  | moto.    | ped.     | cone     | barrier  |
| :--------------: | :------: | :------: | :------: | :------: | :------: | :------: | :------: | :------: | :------: | :------: | :------: |
| ReCo             | 36.1     | 52.2     | 40.9     | 14.3     | 41.8     | 24.2     | 42.8     | 45.9     | 29.5     | 31.2     | 38.3     |
| GLIGEN           | 36.3     | 52.8     | 40.7     | 14.1     | 42.0     | 23.8     | 43.5     | 46.2     | 30.2     | 31.7     | 38.4     |
| ControlNet       | 36.4     | 52.8     | 40.5     | 13.6     | 42.1     | 24.1     | 43.9     | 46.1     | 30.3     | 31.8     | 38.6     |
| **GeoDiffusion** | **38.3** | **53.2** | **43.8** | **18.3** | **45.0** | **27.6** | **45.3** | **46.9** | **30.5** | **32.1** | **39.8** |

▾ Table R3. **Comparison of trainability on COCO.**

| Method           | mAP      | AP50 | AP75 | APm | APl |
| :--------------: | :------: | :-------: | :-------: | :------: | :------: |
| GLIGEN           | 36.8     | 57.6      | 39.9      | 40.3     | 47.9     |
| ControlNet       | 36.9     | 57.8      | 39.6      | 40.4     | 49.0     |
| **GeoDiffusion** | **38.4** | **58.5**  | **42.4**  | **42.1** | **50.3** |

▾ Table R4. **Comparison of inpainting on COCO.**

| Method           | mAP      | AP50 | AP75 |
| :--------------: | :------: | :-------: | :-------: |
| ControlNet       | 17.8     | 25.7      | 20.2      |
| GLIGEN           | 18.3     | 25.8      | 20.9      |
| **GeoDiffusion** | **19.0** | **26.2**  | **21.6**  |

---

### Meta-Review · Area_Chair_igMv · 2023-12-07

**Metareview:**

This work proposes a stable diffusion based technique to generate synthetic datasets for object detection. All the reviewers lean towards accepting the work. Reviewers appreciated the well-written paper with good results on object detection. Multiple reviewers raised concerns with respect to technical novelty w.r.t. some recent works such as GLIGEN as well as some missing experiments. Authors addressed several of these concerns in the rebuttal and reviewers are satisfied with their responses. The reviewers did raise some valuable concerns that should be addressed in the final camera-ready version of the paper, which include adding the relevant rebuttal discussions and revisions in the main paper. The authors are encouraged to make the necessary changes to the best of their ability.

**Justification For Why Not Higher Score:**

The comparisons and novelty w.r.t. existing works is still not solid.

**Justification For Why Not Lower Score:**

All the reviewers are fine with accepting the paper after rebuttal.

---

### Decision · Program_Chairs · 2024-01-16

Accept (poster)